

# Recognition of spatial framework for water quality and its relation with land use/cover types from a new perspective: A case study of Jinghe Oasis in Xinjiang, China

Fei ZHANG [1,2,3*]   Juan WANG [4]   Xiaoping WANG [1,2,3]

*1. College of Resources and Environment Science, Xinjiang University, Urumqi, Xinjiang 830046*

*2. Key Laboratory of Oasis Ecology, Xinjiang University, Urumqi, Xinjiang 830046*

*3. Key Laboratory of Xinjiang wisdom city and environment modeling, Urumqi, Xinjiang 830046*

*4. College of Geography and Remote Sensing Science, Beijing Normal University, 100875, Beijing*

**Abstract:** Quality evaluation for surface water is an important issue in water resource management and protection. To understand the relation between the spatial framework of water quality in Jinghe Oasis and the change in land use/cover type, we first divide 47 water sampling sites measured in May and October 2015 into 6 cluster layers using the self-organizing map (SOM) method based on non-hierarchical *k*-means classification, and then determine the distribution characteristics of water quality from the time sequence. Water quality indices include chemical oxygen demand (COD), biological oxygen demand (BOD), suspended solids (SS), total phosphorus (TP), total nitrogen (TN), ammonia nitrogen ($NH_3-N$), chromaticity (SD), and turbidity (NUT). On the basis of the results, we collect data regarding changes in farmland land, forest-grass land, water body, salinized land, and other land types during the wet and dry seasons and combine these data with the classification results of the GF-1 remote sensing satellite obtained in May and October 2015. We then discuss the influences of land use/cover type on water quality at different layers and seasons. The results indicate that Clusters 1 to 3 provide monitoring samples for the wet season (May 2015), whereas Clusters 4 to 6 provide monitoring samples for the dry season (October 2015). In general, the COD, SS, NUT, TN, and $NH_3-N$ contents are high in Clusters 1and 2. The SD values for Clusters 1, 4, and 6 are high. Moreover, high BOD and TP values are mainly concentrated in Clusters 4 and 6. Through the discussion on the relation between different layers of water quality and land use/cover type change, we determine that the influences of farmland land, forest-grassland, and salinized land are significant on the water quality parameters in Jinghe Oasis. In Clusters 1, 2, and 6, the size of the water area also influences the change in water quality parameters to a certain extent. In addition, the influences of various land use/cover types on the water quality parameters in the research zone during different seasons exhibit the following order: farmland land → forest-grass land → salinized land → water body → others. Moreover, influence is less during the wet season than during the dry season. In conclusion, developing research on the relation between the spatial framework of water quality in Jinghe Oasis and land use/cover type change will be significant for the time sequence distribution of water quality in arid regions from both theoretical and practical perspectives.

**Key words:** SOM; Water quality spatial distribution; Land use/cover; Correlation analysis; GIS

## 0 Introduction

Water quality is of great importance to the study of water resources in arid regions. Accurate information on the spatial distribution of surface water quality is imperative for assessment of environmental monitoring, land surface water management as well as watershed changes (NRC, 2008; Sun et al., 2012). Land use/cover change in drainage basins significantly influences water quality in rivers, lakes, river mouths, and coastal areas (Huang et al.,2013a; Bu et al.,2014; Hur et al.,2014). Surface water

* Corresponding author: Zhangfei   Tel: +86 13579925126   E-mail address: zhangfei3s@163.com

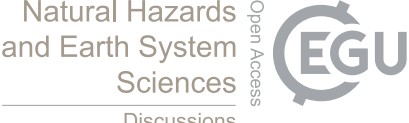

resources, through runoff or infiltration, will always carry a huge amount of pollutants (Swaney et
al.,2012). Therefore, the spatial allocation of land use and land cover change in drainage basins frequently
influences or even poses dangers to water quality through non-point source pollution (Swaney et al., 2012).
However, the regional difference and complexity of land use/cover type result in various corresponding
relations between land use/cover and water quality in different regions (Yang et al., 2016). Therefore,
Therefore, it is very important to explore the relationship between land use/cover types and water quality
for development and management of the basin. (Uuemaa et al., 2005; Xiao et al., 2007; Wan et al., 2014).
At present, numerous scholars have extensively applied statistical methods to indicate mutual relation
between land use/cover change and water quality in various research zones (Céréghino et al.,2009;
Bierman et al.,2011; Huang et al.,2013b). These methods include correlation analysis (Lee et al.,2009; Li et
al.,2015), multiple regression (Park et al.,2014), and redundancy analysis (De et al.,2008; Shen et
al.,2015).

51       A self-organizing map (SOM) is one of the branches of artificial neural network algorithms; it is a
self-organizing and self-learning network visual method that can express multi-dimensional spatial data in
low-dimensional points through non-linear mapping (Kohonen,2001). SOM is an all-purpose classification
tool that can connect samples with variables (Kohonen,2013; Zhou et al.,2016). In recent years, SOM has
become increasingly popular in environmental research because of its capacity to deal with non-linear
relations. Kalteh(2008) and Céréghino(2009) discussed the application of the SOM method in
environmental science, particularly in water resource classification. Chon (2011) evaluated the application
of SOM technology in ecology. The high-dimensional, non-linear, and uncertain features of water quality
monitoring data result in certain complexity during the analysis and evaluation of surface water quality
data. Therefore, data mining and the modern mode recognition method have been introduced to analyze
and explain water quality monitoring data, which can offset the deficiency of the traditional method to a
certain extent(Li et al.,2013). Jinghe Oasis has an oasis and desert, which is a typical mountainous zone in
an arid region and an important part of the northern slope of Tianshan Mountains. Under the influences of
drainage basin climate and human activities, the pollution of the regional ecological environment, which
results from the surrounding agricultural and domestic wastewater around Jinghe Oasis that is directly
discharged or discharged through the river, has become an urgent problem related to sustainable
socioeconomic development in Xinjiang. Therefore, a typical section of Jinghe Oasis in the plain area of
the arid region is selected as the research object. The SOM method is applied to recognize the spatial
distribution of water quality in Jinghe Oasis. On the basis of the result, tentative exploration the relation
between water quality and land use/cover change in different clusters and provide new insights into
controlling, managing, and protecting the ecological environment in the Jinghe Oasis.
72       In the present study, the main objectives of this study were to (1) to analysis the spatial framework of
water quality using the self-organizing map (SOM) method based on non-hierarchical k-means
classification (2) to explore the relationship between water quality parameters and land use/cover types in
different clusters; and (3) to analysis the relationship between water quality parameters and land use/cover
types in different stages.

## 1 Materials and methods

### 1.1 Study area

79       The Jinghe Oasis, is located in the center of Eurasia in the northwest Xinjiang Uygur Autonomous
Region at 44°02′∼45°10′N and 81°46′∼83°51′E. Jinghe Oasis is composed of wetland, desert oasis



vegetation and wildlife, is the national desert ecological reserve. The study area has unique wetland
ecological environment, has been listed as the xinjiang uygur autonomous region "wetland nature reserve".
The region has 385 kinds of desert plants, about 64% of China's vast desert plants. The Jinghe Oasis is
once fed by 12 branch rivers belonging to three major river systems and the major rivers are: Bortala River,
Jing River and Kuytun River, which are mainly rivers connected with the Ebinur Lake. Owing to natural
environmental changes and human activities (i.e., modern oasis agricultural development), many rivers
gradually lost their hydraulic connections with the Ebinur Lake, only Bortala River and Jing River now
supply water to the Ebinur Lake. The climate in the Jinghe Oasis is of typical continental arid climate,
which annual average temperature is 7.36℃, average precipitation is 100~200mm, and average
evaporation is 1500~2000mm (Zhang et al., 2015). In recent years, under the dual drive of the natural and
human factors, Jinghe oasis water resources degraded seriously, outstanding performance in natural oasis
area and water area shrinking, land desertification, salinization of farmland, grassland degradation of
serious, salinization of water quality (Jilil et al.,2002). At the same time, under the effect of strong wind in
Alashankou, the region has become the main source of dust; affect the ecological environment of the area
of northern Xinjiang. This research through the distribution of actual mining spotting, establish research
scope as shown in figure 1.
**Fig.1** Location of the study area

## 1.2 Data acquisition and processing

(1) We applied GF-1 remote sensing images obtained in May and October 2015 as data sources (see
http://www.cresda.com/CN/). These images were not influenced by cloud, fog, and snow cover, and their
quality was good. We conducted radiation and orthographic corrections for the remote sensing image data
combined with 1:50,000 digital elevation model (DEM) data. We established five land use/cover types by
Environment for Visualizing Images software (ENVI Version 5.0), namely, farmland land, forest-grassland,
water body, salinized land, and others, based on the actual conditions of the research zone. Finally, we
generated a vector data map of land use/cover for two stages of the research zones.
(2) The pillar industries in Jinghe Oasis include salt production and *Artemia* breeding. No heavy
industry is present, and thus, point-source pollution from industrial wastewater is not considered in the
research zone. Research samples were collected from the agricultural land in Jinghe County and Tuotuo
Village, which surround Ebinur Lake, a national ecological zone in Ebinur Lake Bird Isle, and the Ganjia
Lake *Haloxylon* natural conservation area. We collected 47 water samples, 23 in May and 24 in October
2015. The monitoring indices used included chemical oxygen demand (COD), five-day biological oxygen
demand (BOD$_5$), suspended solids (SS), total phosphorus (TP), total nitrogen (TN), ammonia nitrogen
(NH$_3$−N), chromaticity (SD), and turbidity (NUT). All polyethylene bottles were used to store the samples.
The bottle was cleaned, dried, and sealed with deionized water before sampling. The sample was taken to
the laboratory for measurement and analysis after collection. We applied dichromate titration, dilution and
inoculation, gravimetry, ammonium molybdate spectrophotometry, alkaline potassium persulfate
decomposition and UV spectrophotometry, and Nessler reagent spectrophotometry to measure COD, BOD$_5$,
SS, TP, TN, and NH$_3$−N, respectively. The analyses of all the samples were entrusted to and completed by
Urumqi Jincheng Measurement Technology Co., Ltd.

## 1.3 Recognition of water quality spatial characteristics based on the SOM method with non-hierarchical *k*-means classification

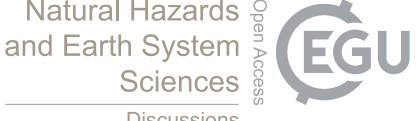

At present, classification based on the SOM neural network is mainly unsupervised and applied to
fault analysis, text clustering, and water quality evaluation. The method, which does not require consistent
data distribution, is simple and can better address detailed information without the influence of minor local
problems. The results are still distribution features for input mode and topological structure (Li et al.,2010).
A typical SOM network generally consists of input and output clusters. All the nerve cells in the input
cluster and the weight vectors in the output cluster are connected and classified as typed data using SOM
via a learning process. Accordingly, the $k$-means algorithm is applied to keep each cluster itself compact
and separate between each cluster as much as possible. The number of clusters is formed according to the
Davies–Bouldin index. The lower value the Davies–Bouldin index is, the better those clusters are
differentiated. Through the K-means cluster analysis combined with the Davies–Bouldin index (DBI) to
select clustering number. The lower value the Davies–Bouldin index is, the better those clusters are
differentiated (Zhou et al., 2016).
The SOM method based on non-hierarchical $k$-means classification was applied to the spatial
framework of water quality in the research zone by implementing the following steps. (1) We typed the
water sample data for clustering from May and October 2015 to the SOM network. We applied the
topological values for calculating network size to select the quantity of nerve cells and determine the
output results based on the minimum values of the Quantity Error (QE) and the Topology Error (TE). QE
was used to determine the capacity of the established neural network in distinguishing the original input
data, whereas TE was used to measure the quality of a neural network, i.e., to evaluate whether the network
would be applicable for training (Kohonen, 2001). After determining network size, we conducted network
training and obtained a set of weight values. (2) The weight value obtained from the SOM cluster results
was considered the initial cluster center, and the $k$-means algorithm was initialized to execute this
algorithm and combined with the DBI index to select clustering number. Such clustering combination
algorithm maintains the self-organizing features of the SOM network, inherits the high efficiency of the
$k$-means algorithm, and offsets the poor clustering effects that result from the excessive convergence time
of the SOM network and the inappropriate selection of the initial clustering center for the $k$-means
algorithm. The SOM requires SOM toolbox and some basic functions to achieve its function in Matrix
Laboratory (MATLAB)(Zhang,2015). In this study, the calculation platform used was MATLAB 2013a.

**1.4 Spatial analysis of the influences of land use/cover change on water quality**

As an artificial system disturbance, land use/cover type is the second major boundary condition that
directly or indirectly influences the hydrologic process and exerts a considerable effect on drainage water
environment. First, we obtained information regarding land use/cover within the scope of the 1km buffer
zone of the water quality sampling point in the research zone using the spatial analysis function of ArcGIS
9.3. On the basis of the result, we then discussed and analyzed the correlation between water quality and
land use/cover type change at different levels and periods. For different levels, we established the
correlation between water quality and land use/cover type in each layer and discussed the influence of land
use/cover type change on water quality. For different periods, we conducted related analysis for land
use/cover information and eight types of water quality indices during the dry and wet seasons. The land
use/cover information and eight types of water quality indices were imported into Canoco4.5 (Ter Braak
and Smilauer, 2002) to test the DCA gradient axis. The results showed that the DCA gradient shaft length
was less than 3. Therefore, the redundancy analysis (RDA) method was applied to determine the influence
trend of land use/cover change within the buffer area in Ebinur Lake on water quality. Such method would
indicate the contribution rate of a single variable of land use/cover on water quality and would directly



demonstrate the correlation between land use/cover type and water quality parameters via a 2D ordination graph. The methodology is explained in the following section and a conceptual flow chart describing the methodology is shown in (Fig.2).

**Fig.2** Conceptual model for the methodology

## 2 Results and analysis

### 2.1 Spatial framework of water quality in Jinghe Oasis

With regard to network structure selection, the neural network with a more complicated structure will generally have better capability to deal with complicated non-linear problems, but will require a longer training time (Kohonen, 2013). Using more water quality indices can provide more abundant information; however, the correlation among indices will increase. The topological values are selected to determine grid size in this study, and the *k*-means clustering method is adopted to obtain results. Overall, after the standard processing of water quality data, the best network training effect is obtained from 35 (7×5) nerve cells; and the QE and TE values are 1.033 and 0.001, respectively.

When the values of average variance are less than 5% in different clusters, the DBI is the lowest, so the corresponding number of clustering can be regarded as the best clustering results. Therefore, this study input trained weights of neuron node, through the K-means cluster analysis combined with the DBI to select clustering number, the results shown in figure 3a. Figure 3a show that six clusters was formed according to the DBI, where minimal value is at six clusters.

**Fig.3** (a) Davies–Bouldin index plot. (b) The results of SOM clustering of the cells on the map plane (Distribution of sampling sites on the SOM according to the eight water quality parameters, and clustering of the trained SOM.)

Figure 3b presents results of SOM clustering of the cells on the map plane, which exhibits similarity among different monitoring stations. In particular, Cluster 1 includes the sampling points around the Ganjia Lake *Haloxylon* natural conservation area in southern Ebinur Lake , east of Ebinur Lake and around Kuitun River during the wet season. Cluster 2 includes the monitoring stations in Jing River and around the agricultural ditch in western Ebinur Lake. Cluster 3 comprises the sampling points of water from melted ice in the southern–western corner of the research zone, which have been grouped into only one type. Cluster 4 includes the sampling points within the Ganjia Lake *Haloxylon* natural conservation area during the dry season. Cluster 5 includes the sampling points in Jing River , the agricultural ditch and around the Ebinur Lake. Cluster 6 is located around Kuitun River and Ebinur Lake Bird Isle, which have more pools. In general, although individual points may interfere with the explanation of the results, the classification results can better identify time sequence features in the research zone. Clusters 1 to 3 provide 100% samples from the wet season (May 2015), whereas Clusters 4 to 6 provides monitoring samples from the dry season (October 2015). To further observe information regarding water quality parameters in Jinghe Oasis through the response of different nerve cells, water quality information from various cluster groups is visualized. The results are shown in Figure 4.

**Fig.4** The patterning results for water quality parameters on the SOM plane

Figure 4 shows the distribution relation among different classifications of various variables and





similar distribution methods among water quality parameters. For example, high COD, TN, NH$_3$−N, and
SD are recorded in the right corner of the SOM network, thereby indicating a declining trend in the
southern Ebinur Lake and the surrounding Kuitun River. The values during the wet season are higher than
those during the dry season. High values of SS and NUT are observed in the left corner of the SOM
network, which indicates the location within the scope of the agricultural ditch and Jing River during the
dry season. This region is mainly distributed around the agricultural land area and is significantly
influenced by human activities. High TP values are observed within the scope of the lower left corner,
which mainly focuses on the agricultural ditch and Jing River during the dry season. The crops are mature
during the dry season, and farmland area is increased, thereby increasing TP value. By contrast, the
distribution of BOD$_5$ is different, which indicates a declining trend from the center to the surrounding area.
High BOD$_5$ values are mainly observed downstream of the Ganjia Lake *Haloxylon* natural conservation
zone, which is surrounded by a salt field, thereby exerting a certain influence on surrounding water quality.
The regional change of water quality in Jinghe Oasis is reflected clearly and directly through SOM. To
observe the distribution of water quality parameters directly, we collect different values of water quality
parameters at various layers (Figure 5).
**Fig.5** Average values for the water quality parameters
Figure 5 shows that the distribution of water quality parameters varies in different clustering layers.
Among the six clusters, water quality is generally relatively better in Cluster 3. However, Cluster 3 only
has one sampling point, which comprises ice and snow water. Therefore, its water quality is not considered.
In addition, COD, SS, NUT, TN, and NH$_3^+$−N contents are high in Clusters 1 and 2, which indicates a
relatively lower water quality compared with the water downstream of the Ganjia Lake *Haloxylon* natural
conservation zone, surrounding Ebinur Lake, and the agricultural land. The SD values are high in Clusters
1, 4, and 6. Meanwhile, high BOD$_5$ values are mainly concentrated in Clusters 4 and 6, around the Ganjia
Lake *Haloxylon* natural conservation zone and Ebinur Lake Bird Isle. Many pools in these clusters are
influenced by the water area to a certain extent. The concentration difference of TP in each layer is
minimal and is considerably influenced by human activities, particularly changes in the agricultural land
area. On the basis of these results and the surface water environment quality standard of China
(GB3838-2002), we evaluate the grades of water quality parameters, including COD, BOD$_5$, TN, NH$_3$−N,
and TP, at different layers (Table 1).
**Table 1** The classes of water quality parameter in each cluster
As shown in Table 1, combined with Chinese Environmental Quality Standard for Surface Water (GB
3838-2002), clusters 1 to 6 do not satisfy potable water level. Clusters 1 and 2 have identical water quality
classification level, and their COD and TN contents are higher than the standard values. In particular, the
COD content exceeds level V. Meanwhile, BOD$_5$ and COD contents are excessive in Clusters 4 and 5,
respectively. In Cluster 6, both COD and BOD$_5$ contents are excessive, but COD content is higher.

**2.2 Analysis of land use/cover type and its relation to water quality at different layers**

From the classification results obtained in May and October 2015 (Figure 6), precision has increased
to 89.9750% and 86.2848%, and the kappa coefficients are 0.8681and 0.8184, respectively, which indicate
an accurate classification result that satisfies the research requirements. Accordingly, ArcGIS is applied as
the water sampling point to establish a 1km buffer zone. The composition of land use/cover at different
layers is analyzed according to the hierarchical results of the water quality parameters, and the results are



presented in Figure 7.
**Fig.6** The change of land use/cover in the Ebinur Lake area during the rainy (May) and dry (October) seasons in 2015(a:

252                              May;b: October)

253                **Fig.7** The area of land use/cover for each cluster

Figure 7 shows land use/cover mode at different layers. In general, the salinization phenomenon is
serious in the entire research zone. Among the six clusters, Cluster 1 mainly includes the sampling points
around the Ganjia Lake *Haloxylon* natural conservation area, eastern Ebinur Lake, and Kuitun River. The
major land types in this cluster are forest-grass land. The monitoring site in Cluster 2 is located in the
irrigation ditch of Jinghe Oasis and in Jing River. Crops do not grow abundantly in May in the research
zone, and the major land types in this cluster are forest-grass land. Cluster 4 mainly includes the Ganjia
Lake *Haloxylon* natural conservation area and the sampling points in Tuotuo Village during dry season,
which are farmland, forest-grassland. The sampling points in Cluster 5 are located in the agricultural ditch,
Jing River, and the surrounding Ebinur Lake. The land types mainly include forest-grassland and farmland,
which is larger than the other land type. Cluster 6 is mainly located in Kuitun River and the surrounding
Ebinur Lake Bird Isle. A considerable number of plants, such as reed, grow in some pools. The percentages
of forest-grass land and salinized land within the 1km scope are large based on the actual conditions in the
research zone and the distribution conditions of the sampling points. On the basis of these results, the
correlation between land use/cover type and water quality parameters from Clusters 1 to 6 is analyzed. The
results are presented in Table 2.

269                **Table 2** The correlation coefficients between land use/cover and water quality parameters in each cluster

In Cluster 1, forest–grass land exhibit a negative correlation with SS and NUT under the significance
level of 0.01, with coefficients reaching up to −0.710 and −0.724, respectively. At a significance level of
0.05, water body exhibits an obvious positive correlation with COD and a negative correlation with TN,
with coefficients of 0.986 and −0.721 respectively. At a confidence level of 0.01, salinized land
demonstrates a positive correlation with NUT, with a coefficient of 0.756. In Cluster 2, farmland presents a
negative correlation with COD and a positive correlation with NUT at a confidence level of 0.05, and the
coefficients are −0.581 and 0.639, respectively. Under the same condition, forest–grass land exhibit a
positive correlation with COD, and the coefficient is 0.613. At a confidence level of 0.01, the water body
exhibits an evident negative correlation with SS and NUT, and the correlation coefficients are −0.983 and
−0.990, respectively. In Cluster 4, several water quality parameters are mainly influenced by farmland,
forest-grass land, and salinized land. At a confidence level of 0.05, the farmland exhibits a negative
correlation with COD, and with a coefficient of −0.652. At a confidence level of 0.01, farmland
demonstrates an evident positive correlation with TP, and the coefficient is 0.872. At a confidence level of
0.05, salinized land shows a clear negative correlation with TP and NUT, and the coefficients are −0.791
and −0.819, respectively. In this layer, others land type exhibit a positive correlation with TP with a
coefficient of 0.868. The sampling point in this layer is mainly located around Tuotuo Village, where the
influences of human activities are considerable; therefore, the correlation percentage of others land type in
Cluster 4 with TP is high. In Cluster 5, farmland demonstrates an evident negative correlation with BOD$_5$
at a 0.01 confidence level, with a correlation coefficient reaching up to −0.881. At a confidence level of
0.05, salinized land shows a positive correlation with BOD$_5$, with a correlation coefficient of 0.774. In
Cluster 6, forest–grass land show a clear negative correlation at a confidence level of 0.01, and the
correlation coefficient reaches −0.884. At a confidence level of 0.05, the water area exhibits a positive
correlation with a correlation coefficient of 0.980.



From the results of the comprehensive analysis, the influences of farmland, forest–grass land, and salinized land are considerable on the water quality parameters in Jinghe Oasis. In Clusters 1, 2, and 6, the size of the water area also influences change in water quality parameters to a certain extent. Given the unbalanced distribution of sampling points at different layers, the effect of land use/cover composition on water quality in the research zone varies, and can indicate influence only to a certain extent. Therefore, considering the actual conditions in Ebinur Lake, different land use/cover types and water quality influences are understood as a whole, and the correlation between land use/cover type and water quality in Jinghe Oasis at different periods is further discussed.

## 2.3 Analysis of land use/cover change in Jinghe Oasis and its correlation with water quality at different seasons

The constituents of land use/cover type at different seasons exert diverse influences on water quality. Therefore, analyze the influences of land use/cover type change on water quality. The results are presented in Figure 8.

**Fig.8** RDA analyses of comprehensive Land use/cover and water quality (a: Wet season;b: Dry season)

As shown in Figure 8, farmland exhibits a negative correlation with COD at a confidence level of 0.01 during the wet season, and the correlation coefficient is −0.543. By contrast, it shows a positive correlation with NUT, and with a correlation coefficient of 0.555. At a confidence level of 0.05, farmland demonstrates a negative correlation with $NH_3-N$, and the correlation coefficient is −0.461. At a confidence level of 0.05, forest-grass land show a positive correlation with $BOD_5$ and TP, with correlation coefficients of 0.470 and 0.518, respectively. They exhibit a negative correlation with SS and NUT, with correlation coefficients of −0.529 and −0.498, respectively. At a confidence level of 0.05, salinized land demonstrates a positive correlation with $BOD_5$ and TP, with correlation coefficients of −0.503 and 0.518, respectively. By contrast, it presents a negative correlation with SS and NUT, with correlation coefficients of 0.449 and 0.449, respectively. During the dry season, the influence of farmland on various water quality parameters evidently increases because of crop growth. At a confidence level of 0.01, farmland presents a clear negative correlation with COD and an evident positive correlation with TP, with correlation coefficients of −0.620 and 0.616, respectively. At a confidence level of 0.05, farmland shows a positive correlation with TN and a negative correlation with $BOD_5$, $NH_3-N$ and SD, with correlation coefficients of 0.543, −0.495, −0.522, and −0.526, respectively. At a confidence level of 0.01, salinized land demonstrates a negative correlation with NUT and TP, and the correlation coefficients are −0.543 and −0.603, respectively. By contrast, it presents a positive correlation with $BOD_5$ at a confidence level of 0.05, and the correlation coefficient is 0.522. Similarly, during the wet and dry seasons, the correlation of the water body and others land type with water quality parameters is small. From the results, the influences of various land use/cover types in the research zone on water quality parameters exhibit the following order: farmland → forest–grass land →salinized land→ water body → others. Moreover, the influence is less during the wet season than during the dry season.

## 3 Discussion and conclusions

### 3.1 Discussion

Given seasonal differences, the unbalanced distribution of precipitation amount results in an apparent variation in surface runoff and further imbalance in the spatial distribution of water quality in the research



zone(Fan et al.,2012; Prathumratana et al.,2008; Li et al.,2015). During the wet season (May) in Jinghe
Oasis, melted water from mountain ice and snow is collected, which promotes flow in Jing River, thereby
resulting in a significant increase in surface runoff and lead to the water quality in rainy season is better
than dry season. During the dry season, the aquatic plants in rivers and lakes are growing with the
temperature rises, which can absorb and purify part of the water quality parameters in a certain degree.
Therefore, a significant change in surface runoff and seasonal change in the research zone are important
factors that result in noticeable differences in the spatial distribution characteristics of water quality during
the wet and dry seasons. Another major factor that results in differences in the spatial distribution of water
quality is land use/cover change, especially the farmland. During the dry season, farmland have great
influenced on more water quality variables than during the rainy season because of intensive fertilization
and agricultural runoff from soil erosion (Ngoye et al, 2004; ;Li et al., 2009; Tran et al., 2010). Multiple
factors threaten the ecological safety of the Jinghe Oasis system. Especially, in recent years, the lakeside
desertification zone has rapidly expanded because of the decrease in the area of Ebinur Lake as well as the
degradation of lakeside vegetation under the deflection of strong winds in Alashankou. In the current
overall situation, the influences of human activities on land use/cover are directly related to the
development of a vulnerable ecological area that surrounds Ebinur Lake.
Recent statistics indicate that the annual growth rate of the population in Jinghe Oasis is
approximately 2.49%, which is slightly higher compared with the previous growth rates (Li, 2006). Under
the stress of a large population, common inappropriate phenomena that occur with land use/cover in Jinghe
Oasis will increase. For the last 30 years, cotton has been the major crop in Jinghe Oasis. The results of the
current study indicate that the research zone is distributed across a farmland, where water quality in the
surrounding sampling points is lower than those in other regions. The primary sources of living of urban
residents around the Ebinur Lake area are agriculture and animal husbandry. Pollutants that result in high
TP and $NH_3-N$ contents in water include excessively applied chemical fertilizers in farmland, livestock
manure in rural villages, randomly stocked garbage, and domestic wastewater. In particular, a vast area
with improperly applied chemical fertilizers and pesticides leads to high nitrogen and phosphorus contents
in water, which result in the spread of algae in some sections of a river. Consequently, the amount of
dissolved oxygen in water is decreased, water quality deteriorates, and eutrophication occurs. Furthermore,
a serious salinization phenomenon exists. Certain measures have been implemented for the ecological
protection of Ebinur Lake, such as returning farmland to forest, cultivating ecological forest, promoting
efficient irrigation and water-saving technology. However, these measures promote the gradual expansion
of the lake area and also results in different degrees of negative consequences. The most apparent result is
the rise of the underground water level, which has aggravated land salinization in lowland areas and has
resulted in vast expanses of uncultivated former agricultural lands. Statistics indicate that soil salinization
in the Ebinur Lake area mainly occur in Bortala River, Jing River, the surrounding villages and towns of
Ebinur Lake, downstream of Daheyanzi River, and north of Bole City (Mi et al.,2010). Severe soil
salinization has seriously affected the farming of crops; therefore, some farmers increase the amount of
chemical fertilizer to increase yield, which increased the pollution of water and soil. Others even abandon
the land, thereby resulting in land use/cover change.
Most rivers in Xinjiang are characterized by low water yield, short flow, small water environmental
capacity, poor self-cleaning capability, and low tolerance to pollution. Hence, an artificial change in land
use and exploration for resources in lake regions lead to an evident correlation between land use/cover type
and water quality. In addition, scientifically utilizing and protecting the water resources of Ebinur Lake, as

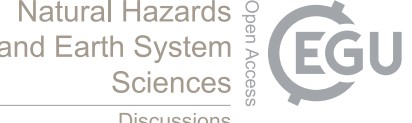



well as scientifically applying chemical fertilizers and improving their application rates, are important issues that should be addressed to achieve sustainable development in the agricultural irrigation zones in Jinghe Oasis and rivers in Xinjiang.

**3.2 Conclusions**

The spatial distribution characteristics of water quality in Jinghe Oasis and their correlation with land use/cover type are analyzed, and the following conclusions are drawn.

(1) Through the SOM method based on non-hierarchical *k*-means classification, 47 sampling points of water quality are divided into 6 types, and time sequence characteristics in the research zone are better recognized in the classification results. Clusters 1 to 3 comprise samples from the wet season (May 2015), whereas Clusters 4 to 6 comprise monitoring samples from the dry season (October 2015). In general, COD, SS, NUT, TN, and $NH_3-N$ contents are high. The SD value is high in Clusters 1, 4, and 6. In addition, the high BOD and TP values are mainly concentrated in Clusters 4 and 6. On the basis of these findings, water quality at different layers of the research zone is further evaluated. The results show that Clusters 1 to 6 do not satisfy potable water level.

(2) The correlation between land use/cover type and water quality parameters from Clusters 1 to 6 is analyzed according to the hierarchical results of the water quality parameters. The comprehensive analysis indicates that the influences of arable land, forest and grassland, and salt lick are significant on the water quality parameters in Jinghe Oasis. In Clusters 1, 2, and 6, the size of the water area also influences changes in water quality parameters to a certain extent.

(3) During the wet and dry seasons, the influences of various land use/cover types in the research zone on water quality parameters exhibits the following order: arable land → forest and grassland → salt lick → water area → others. Moreover, influence is less during the wet season than during the dry season.

In general, land use/cover type, area percentage, and water quality in Jinghe Oasis demonstrate an apparent correlation. The research results can tentative exploration the relationship between water quality and land use/cover types in different clusters by SOM. It provides a new insight for further studies on the correlation between land use/cover and water quality in Jinghe Oasis, as well as a scientific reference for formulating regulation and control policies for the spatial development and water environment protection of the Jinghe Oasis.

**Acknowledgements:** Thanks to the National Meteorological Information Center data provided meteorological data. The research was carried out with the financial support provided by the Natural Science Foundation of Xinjiang Uygur Autonomous Region, China (2016D01C029); National Natural Science Foundation of China (41361045), the Scientific and technological talent training program of Xinjiang Uygur Autonomous Region (grant No. QN2016JQ0041).The authors wish to thank the referees for providing helpful suggestions to improve this manuscript.

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





Figures caption

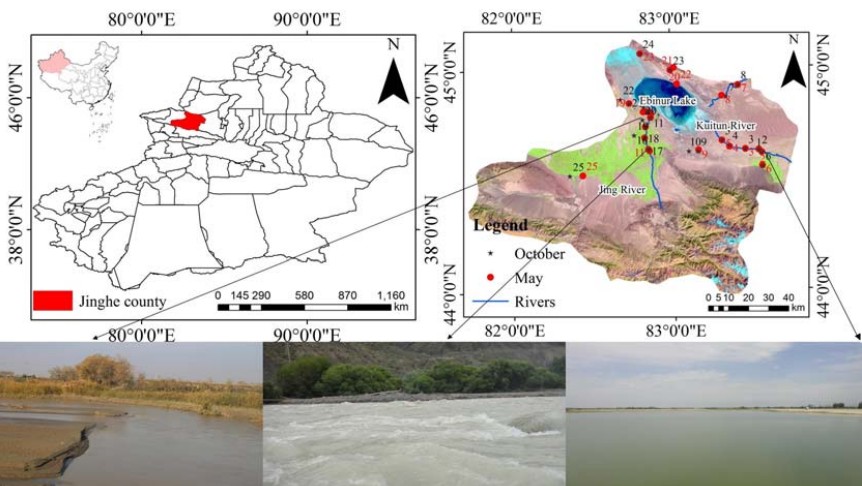

**Fig.1** Location of the study area

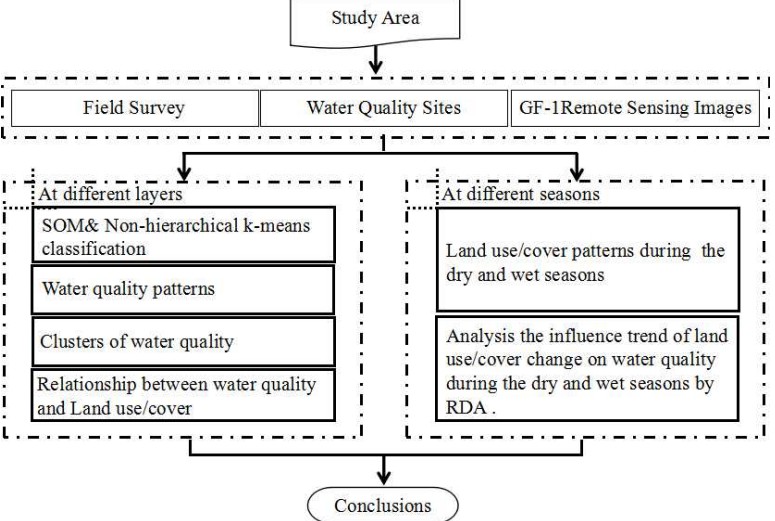

**Fig.2** Conceptual model for the methodology

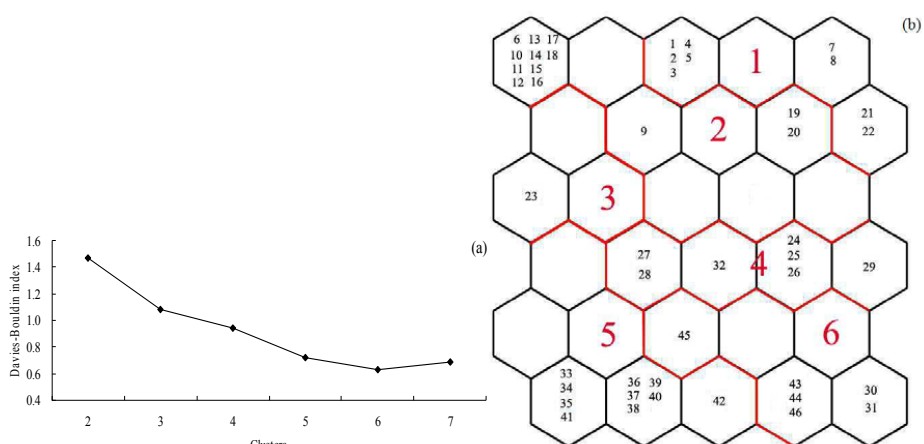

Fig.3 (a) Davies–Bouldin index plot. (b) The results of SOM clustering of the cells on the map plane (Distribution of sampling sites on the SOM according to the eight water quality parameters, and clustering of the trained SOM.)

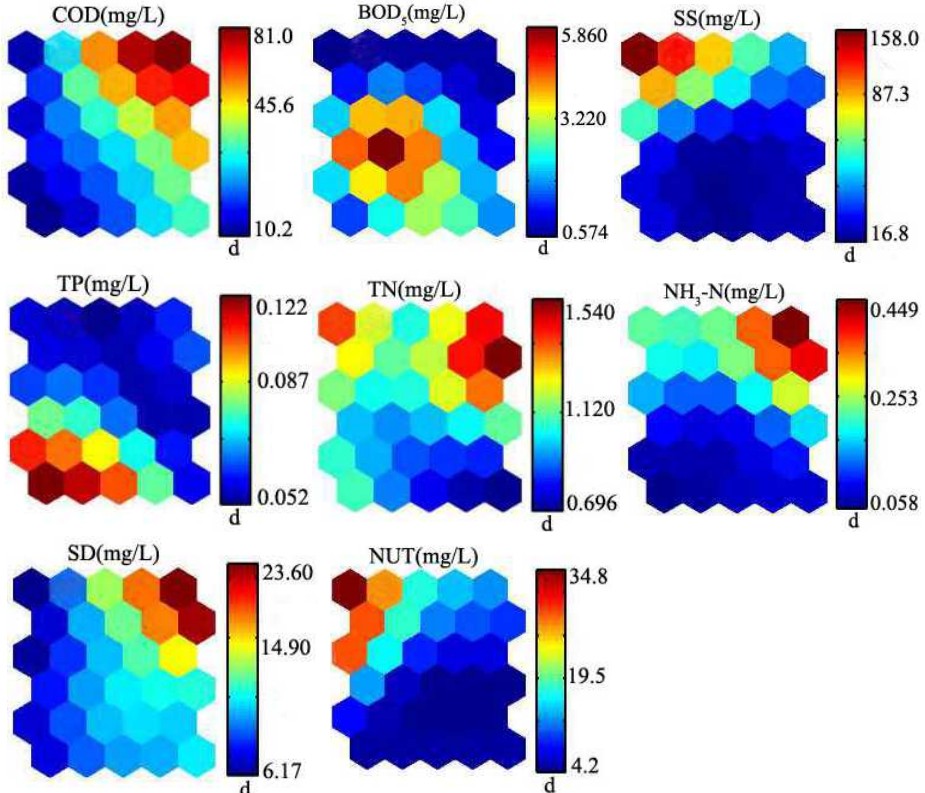

Fig.4 The patterning results for water quality parameters on the SOM plane



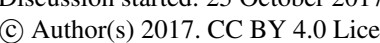

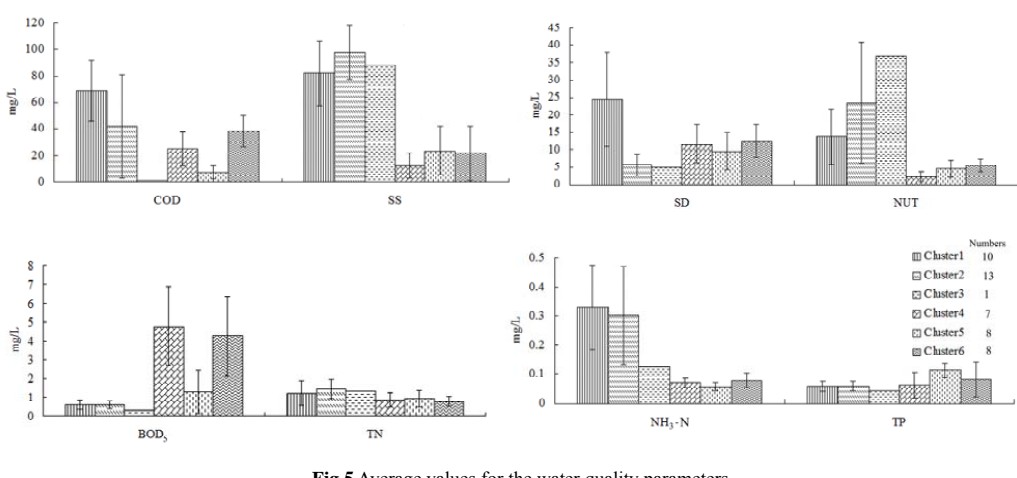

**Fig.5** Average values for the water quality parameters

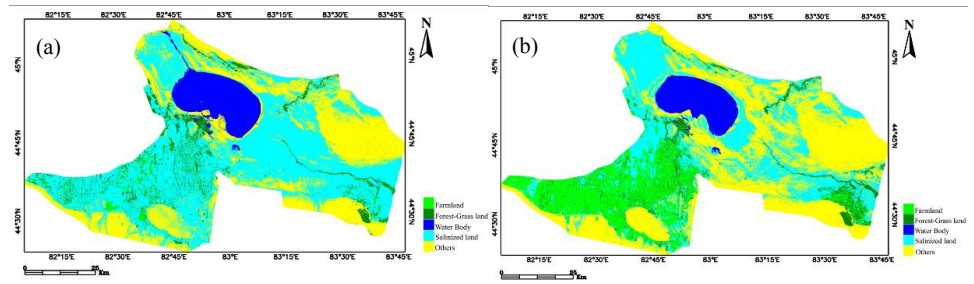

**Fig.6** The change of land use/cover in the Ebinur Lake area during the rainy (May) and dry (October) seasons in 2015(a: May;b:
October)

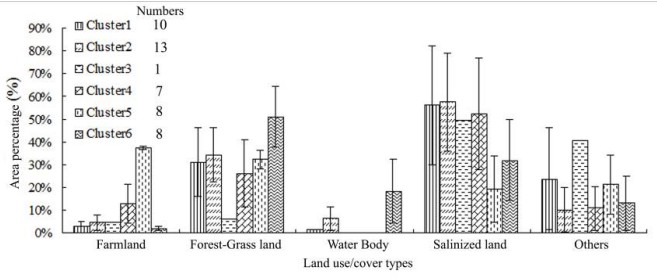

**Fig.7** The area of land use/cover for each cluster



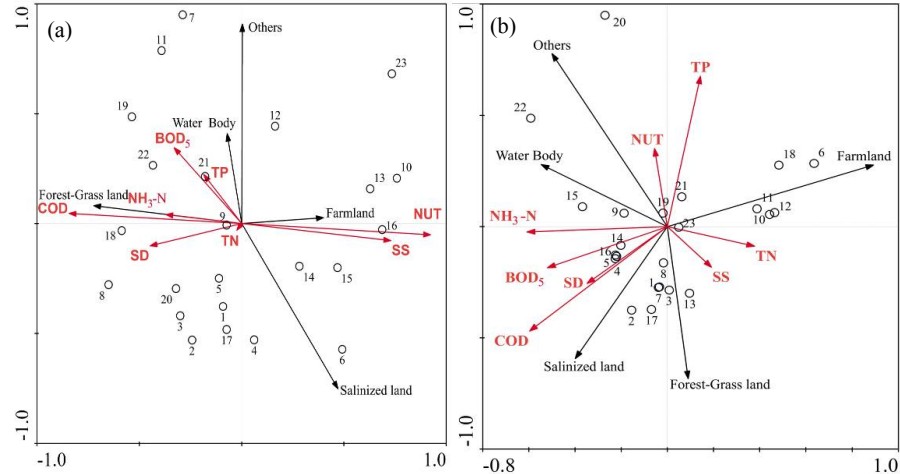

**Fig.8** RDA analyses of comprehensive Land use/cover and water quality (a: Wet season;b: Dry season)

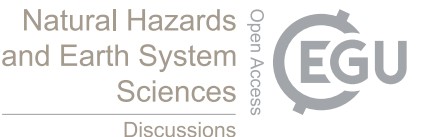
Tables caption

Table 1 The classes of water quality parameter in each cluster

|  | Cluster1 | Cluster2 | Cluster4 | Cluster5 | Cluster6 |
|---|---|---|---|---|---|
| COD | Exceed V | Exceed V | I | IV | V |
| $BOD_5$ | I | I | IV | I | IV |
| TN | IV | IV | III | III | III |
| $NH_3^+$-N | II | II | I | I | I |
| TP | II | II | II | III | II |

Table 2 The correlation coefficients between land use/cover and water quality parameters in each cluster

|  | Parameters | Farmland | Forest-Grass land | Water Body | Salinized land | Others |
|---|---|---|---|---|---|---|
| Cluster1 | COD | -0.161 | 0.240 | **0.986**[*] | -0.110 | -0.361 |
|  | $BOD_5$ | 0.074 | 0.492 | -0.439 | -0.613 | 0.552 |
|  | SS | -0.271 | **-0.710**[**] | 0.801 | 0.619 | -0.384 |
|  | TP | -0.195 | 0.453 | 0.371 | 0.444 | 0.623 |
|  | TN | 0.464 | 0.524 | **-0.721**[*] | -0.224 | 0.121 |
|  | $NH_3$-N | -0.491 | 0.039 | 0.071 | -0.066 | 0.291 |
|  | SD | -0.296 | 0.448 | 0.415 | -0.426 | 0.396 |
|  | NUT | -0.261 | **-0.724**[**] | 0.550 | **0.756**[**] | -0.612 |
| Cluster2 | COD | **-0.581**[*] | **0.613**[*] | 0.916 | **-0.693**[**] | 0.442 |
|  | $BOD_5$ | -0.004 | 0.455 | 0.055 | -0.545 | 0.242 |
|  | SS | 0.493 | -0.512 | **-0.983**[**] | 0.386 | 0.047 |
|  | TP | -0.222 | 0.531 | 0.850 | -0.129 | 0.382 |
|  | TN | 0.351 | 0.415 | -0.867 | -0.356 | -0.311 |
|  | $NH_3$-N | -0.467 | 0.121 | 0.122 | -0.269 | 0.284 |
|  | SD | -0.226 | -0.073 | -0.051 | -0.217 | 0.473 |
|  | NUT | **0.639**[*] | -0.446 | **-0.990**[**] | 0.513 | -0.236 |
| Cluster4 | COD | **-0.652**[*] | 0.484 | / | 0.375 | -0.048 |
|  | $BOD_5$ | -0.482 | -0.402 | / | 0.505 | 0.688 |
|  | SS | -0.155 | 0.658 | / | -0.179 | -0.167 |
|  | TP | **0.872**[**] | -0.398 | / | **-0.791**[*] | **0.868**[*] |
|  | TN | 0.336 | 0.468 | / | -0.571 | -0.124 |
|  | $NH_3$-N | -0.202 | -0.540 | / | 0.398 | 0.352 |
|  | SD | -0.543 | **0.825**[*] | / | 0.214 | -0.549 |
|  | NUT | 0.578 | 0.469 | / | **-0.819**[*] | -0.129 |
| Cluster5 | COD | 0.094 | -0.372 | / | 0.325 | -0.400 |
|  | $BOD_5$ | **-0.881**[**] | 0.503 | / | **0.774**[*] | -0.044 |
|  | SS | 0.621 | -0.533 | / | -0.284 | -0.380 |
|  | TP | 0.587 | -0.702 | / | -0.565 | 0.136 |
|  | TN | 0.735 | -0.588 | / | -0.184 | -0.604 |
|  | $NH_3$-N | -0.675 | 0.108 | / | 0.487 | 0.308 |
|  | SD | -0.632 | 0.330 | / | 0.208 | 0.576 |
|  | NUT | 0.311 | 0.076 | / | -0.459 | 0.154 |
| Cluster6 | COD | 0.489 | 0.401 | **0.980**[*] | -0.454 | 0.289 |
|  | $BOD_5$ | -0.256 | **-0.884**[**] | -0.660 | 0.367 | 0.327 |

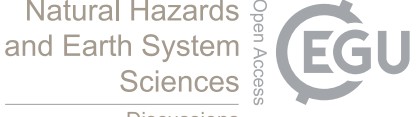



| | | | | | |
|---|---|---|---|---|---|
| SS | -0.481 | 0.194 | 0.341 | -0.150 | -0.062 |
| TP | -0.545 | -0.656 | 0.269 | -0.060 | 0.206 |
| TN | -0.158 | -0.364 | -0.022 | 0.516 | 0.313 |
| NH$_3$-N | -0.553 | -0.366 | 0.517 | -0.090 | 0.603 |
| SD | 0.811 | -0.037 | -0.857 | 0.249 | -0.282 |
| NUT | 0.450 | 0.165 | 0.764 | -0.497 | 0.636 |

$^*$p<0.05(2-tailed)    $^{**}$ p<0.01(2-tailed)