# Peer review of "Recognition of spatial framework for water quality and its relation"

_Natural Hazards and Earth System Sciences, 2017_

## Referee Comment (RC1) · Anonymous Referee #1 · 10 Nov 2017

This manuscript concentrates on identifying important spatial drivers for stream water quality in China. The authors try to explain the key drivers of water quality by clustering the data by using self-organizing map method. In my opinion, the approach and method used may be valuable to be published. However, I have several concerns, which can be grouped into three aspects: 1) The abstract should not only include the methodology, but also the novelty of this paper, so the authors should add innovation clearly in the Abstract. 2) It is noted that your manuscript needs careful editing by someone with expertise in technical English editing, paying particular attention to English grammar,

spelling, and sentence structure so that the goals and results of the study are clear to the reader. 3) In my opinion, maybe the "Discussions" as an independent part is better. Also, the authors should improve the Discussions in the section "conclusions and discussions" The subject is interesting and meaningful. I think it can be published after minor revision.

---

## Short Comment (SC1) · 10 Nov 2017

This is a solid work that combines field sampling and statistical analysis to reveal the correlation between water quality and LULC types. While the experiments have been designed properly and the results been presented reasonably, I have some specific comments to this manuscript for discussion. 1. "Spatial framework" in the title and in the content sounds a little bit weird, maybe replaced with spatial pattern or spatial distribution? 2. Line 132, repeated sentence. 3. Figure 3, it seems there is only one sample in cluster 3? How much significance would the results of this cluster have?

Should this cluster be deleted and keep only 5 clusters? 4. Table 1, it seems there is no difference between cluster 1 and cluster 2? Then, what makes them two clusters? 5. Table 2, it would be better to add lines to separate each clusters. 6. Line 239, "combined with" should be "according to". 7. Line 245-246, how was the accuracy derived for land cover maps in Figure 6?

---

## Referee Comment (RC2) · Anonymous Referee #2 · 10 Dec 2017

This manuscript conducted a very detailed work on investigating the water quality and its relation with land use/cover types. While the data and samples have been processed carefully and rigorously, the results look reasonable and trustful. However, I have several major concerns regarding to this work.

There are two key words in the title that I cannot find strong evidence from the context to support, the "spatial" and "a new perspective". I cannot see how the authors make their findings 'spatial', since there is nothing that has been shown as a map, except for the study area (Figure 1) and land cover (Figure 6). From the Introduction, I also cannot

see why the authors labeled their work as "a new perspective"? It is new because of what? This should be made clear in the Introduction.

I am also unclear about the role that remote sensing data played in this study. What was it for, only for revealing the land use/cover types of each sample? How was it used? It was said there was a 1km buffer zone established for each sample. Then how? And why 1km? Besides, in line 246, there is the accuracy of land use/cover classification. I am wondering how these accuracy numbers were derived? Using what as reference (ground truth)?

The grammar and usage have pervasive problems. It needs to be comprehensively edited by someone with strong English language skills before publication. There are some words, such as 'framework' in the title, and 'layers' in line 19, that are very confused to me. I have no idea what they are referring to.

Line 132, Davies-Bouldin index just comes out suddenly. What is this? What is the reference for it? What do you mean by "Through the K-means . . ."? These content needs to be rewritten.

What is the DCA gradient axis in line 163? Why do you say "Therefore, the redundancy analysis (RDA) method was applied to . . .". I cannot see connections between the sentences before the word 'Therefore' and those after it.

Sentence in line184-185 needs to be rewritten.

---

## Author Comment (AC1) · 11 Jan 2018

**Replies to the comments and suggestions**

**Dear editor and C. Huang:**

Thank you for your letter and for the reviewers' comments concerning our manuscript entitled" Recognition of spatial framework for water quality and its relation with land use/cover types from a new perspective: A case study of Jinghe Oasis in Xinjiang, China". Those comments are all valuable and very helpful for revising and improving our paper, as well as important guiding sense to our researches. The authors have studied comments carefully and have made correction which we hope meet with approval. Revised portions are marked in red in the paper and authors have tried our best to revise the manuscript based on their suggestions. The point-by-point responses to each of the comments are presented as follows.

1. "Spatial framework" in the title and in the content sounds a little bit weird, maybe replaced with spatial pattern or spatial distribution?

**Reply:** Thanks to Revivers for encouraging Authors to revise the manuscript. Thanks to Reviewers for their suggestions in improving the manuscript. Authors have taken reviewers' comments and suggestions seriously. The title is revised, "Spatial framework" change into "spatial pattern". Modified parts have been marked in red in the revised paper.

2. Line 132, repeated sentence.

**Reply:** Line 132, repeated sentence has deleted by the authors.

3. Fig. 3, it seems there is only one sample in cluster 3? How much significance would the results of this cluster have? Should this cluster be deleted and keep only 5 clusters?

**Reply:** Only one sample of cluster 3 cannot be deleted, because the results are automatic output by SOM clustering. The results show the water quality sample is special of water quality. For significance of the cluster results, the SOM technique is a powerful tool to group the similar input patterns from a multidimensional input space into a much lower dimensional space, usually two dimensions; SOM can be used for clustering, classification, estimation, prediction, and data mining (Yan et al., 2016; Park et al., 2014), the spatial distribution of water quality is classified by SOM technique. In the present study, the main objectives of this study were analysis the relationship between water quality parameters and land use/cover types in different stages. The water samples are divided into six categories, which references the results of previous studies and consider of the status

of water quality (Park et al., 2014; Ren et al., 2017).

**Reference**

Yan, A., Zou, Z., & Li, R. 2016. Descriptive characteristics of surface water quality in hong kong by a self-organising map. International Journal of Environmental Research & Public Health, 13(1):115.

Park, Y. S., Kwon, Y. S., Hwang, S. J., & Park, S. 2014. Characterizing effects of landscape and morphometric factors on water quality of reservoirs using a self-organizing map. Environmental Modelling & Software, 55(5):214-221.

Ren Y, Zhang F, Wang J, Zhang Y, Yang ST. 2017. Spatio-temporal characteristics and source identification of surface water pollutants in Lake Ebinur Watershed,Xinjiang. Journal of Lake Sciences, 2017, 29(5):1143-1157.

4. Table 1, it seems there is no difference between cluster 1 and cluster 2? Then, what makes them two clusters?

**Reply:** Thanks to Reviewers for their suggestions in improving the manuscript. Table 1 shows the results of water quality combined with Chinese Environmental Quality Standard for Surface Water (GB 3838-2002), clusters 1 to 6 shows different water quality standard. Clusters 1 and 2 have identical water quality classification level, and their COD and TN contents are higher than the standard values. The result of this classification is not a fixed value, but an interval value. the SOM technique is a powerful tool to group the similar input patterns from a multidimensional, usually two dimensions; SOM can be used for clustering, classification, estimation, prediction, and data mining (Yan et al., 2016; Park et al., 2014), the methods can mining more detailed spatial distribution patterns, there is difference between cluster 1 and cluster 2 in more detailed spatial distribution patterns.

**Reference**

Yan, A., Zou, Z., & Li, R. 2016. Descriptive characteristics of surface water quality in hong kong by a self-organising map. International Journal of Environmental Research & Public Health, 13(1):115.

Park, Y. S., Kwon, Y. S., Hwang, S. J., & Park, S. 2014. Characterizing effects of landscape and morphometric factors on water quality of reservoirs using a self-organizing map. Environmental Modelling & Software, 55(5):214-221.

5. Table 2, it would be better to add lines to separate each clusters.

**Reply:** Thanks to Reviewers for their suggestions in improving the manuscript. According to Reviewers for their suggestions, lines are added in the Table 2. Modified parts have been marked in red in the revised paper. Please see the article.

6. Line 239, "combined with" should be "according to".

**Reply:** Thanks to Reviewers for their suggestions in improving the manuscript. "Combined with" changed into

"according to". Modified parts have been marked in red in the revised paper. Please see the article on page 6, lines

to 239 in revised manuscript with obviously marked.

7. Line 245-246, how was the accuracy derived for land cover maps in Figure 6?

**Reply:** The accuracy classification for land cover maps in Figure 6 is required by Land cover transition matrix.

Confusion matrix is added by the authors, Please see the article, as follows:

Table 2 The calculation of a confusion matrix by a maximum likelihood supervised classification

| | LULC | Water body | Saline land | Farmland | Forest grassland | Other land | Total | User's accuracy (%) |
|---|---|---|---|---|---|---|---|---|
| | Water body | 144 | 0 | 0 | 0 | 0 | 144 | 100 |
| | Saline land | 0 | 77 | 0 | 0 | 16 | 93 | 82.79 |
| | Farmland | 0 | 36 | 101 | 0 | 0 | 137 | 73.72 |
| May | Forest-Grass land | 0 | 36 | 0 | 101 | 0 | 137 | 73.72 |
| | Other land types | 1 | 0 | 0 | 0 | 87 | 88 | 98.96 |
| | Total | 145 | 149 | 101 | 101 | 103 | Overall=89.9750% | |
| | Producer's accuracy (%) | 99.31 | 51.67 | 100 | 100 | 84.46 | Kappa=0.8681 | |
| | Water body | 144 | 0 | 0 | 0 | 0 | 144 | 100 |
| | Saline land | 0 | 57 | 0 | 0 | 26 | 83 | 86.67 |
| | Farmland | 0 | 16 | 101 | 0 | 0 | 117 | 86.32 |
| October | Forest-Grass land | 4 | 16 | 0 | 101 | 0 | 117 | 86.32 |
| | Other land types | 0 | 0 | 0 | 0 | 77 | 77 | 100 |
| | Total | 148 | 89 | 101 | 101 | 103 | Overall=86.2848% | |
| | Producer's accuracy (%) | 97.29 | 64 | 100 | 100 | 74.75 | Kappa=0.8184 | |

In addition, authors have revised the figures and tables as well as words expression in the entire manuscript, please see the text.

Authors tried our best to improve the manuscript and made some changes in the manuscript. Authors appreciate for Editors/Reviewers' warm work earnestly, and hope that the correction will meet with approval. Once again, thank you very much for your comments and suggestions.

All in all, if you have any questions about our paper, please contact with me as follow address:

E-mail:zhangfei3s@163.com

Thanks very much.

Best wishes and warmly regards for you.

Sincerely yours Fei ZHANG

11[th], Jan., 2017

---

## Author Comment (AC2) · 11 Jan 2018

**Replies to the comments and suggestions**

**Dear editor and Anonymous Referee #1:**

Thank you for your letter and for the reviewers' comments concerning our manuscript entitled" Recognition of spatial framework for water quality and its relation with land use/cover types from a new perspective: A case study of Jinghe Oasis in Xinjiang, China". Those comments are all valuable and very helpful for revising and improving our paper, as well as important guiding sense to our researches. The authors have studied comments carefully and have made correction which we hope meet with approval. Revised portions are marked in red in the paper and authors have tried our best to revise the manuscript based on their suggestions. The point-by-point responses to each of the comments are presented as follows.

**Anonymous Referee #1**

**Q:** The abstract should not only include the methodology, but also the novelty of this paper, so the authors should add innovation clearly in the Abstract.

**Reply:** Thanks to Reviewers for their suggestions in improving the manuscript. Authors have taken reviewers' comments and suggestions seriously. The abstract is revised by authors. Modified parts have been marked in red in the revised paper. Please see the article on page 1, lines to 9-29 in revised manuscript with obviously marked. Please see as follows:

To understand the relation between spatial water quality patterns and changes in land use/cover types in the Jinghe Oasis, this study divided 47 water sampling sites, which were measured in May and October 2015, into 6 cluster layers using the self-organizing map (SOM) method based on non-hierarchical $k$-means classification. Next, it determined the distribution characteristics of water quality indices for the time sequence. The water quality indices included the chemical oxygen demand (COD), biological oxygen demand (BOD), suspended solids (SS), total phosphorus (TP), total nitrogen (TN), ammonia nitrogen ($NH_3-N$), chromaticity (SD), and turbidity (NUT). Based on the results, we collected data on the changes in the farmland, forest-grass land, water body, salinized land, and other land types during the wet and dry seasons and combined these data with the classification results of the GF-1 remote sensing satellite data obtained in May and October 2015. We then discussed the influences of land use/cover type on water quality for different layers and seasons. The results indicate that Clusters 1 to 3 included monitoring samples from the wet season (May 2015), whereas Clusters 4 to 6 included monitoring samples from the dry season (October 2015). In general, the COD, SS, NUT, TN, and $NH_3-N$ values were high in Clusters 1 and 2. The SD values for Clusters 1, 4, and 6 were high. Moreover, high BOD and TP values were mainly concentrated in Clusters 4 and 6. In the discussion on the relation between the different water quality

parameters and land use/cover type changes, we determined that farmland, forest-grassland, and salinized land significantly influenced the water quality parameters in the Jinghe Oasis. In Clusters 1, 2, and 6, the size of the water area, to a certain extent, also influenced changes in the water quality parameters. In addition, the influences of various land use/cover types on the water quality parameters in the research zone during the different seasons exhibited the following descending order of magnitude: farmland → forest-grass land → salinized land → water body → others. Moreover, their influences were lower during the wet season than the dry season. In conclusion, developing research on the relation between the spatial framework of the water quality in the Jinghe Oasis and land use/cover type changes is significant for the time sequence distribution of water quality in arid regions from both theoretical and practical perspectives.

**Q:** It is noted that your manuscript needs careful editing by someone with expertise in technical English editing, paying particular attention to English grammar, spelling, and sentence structure so that the goals and results of the study are clear to the reader.

**Reply:** Thanks to Revivers for encouraging Authors to revise the manuscript. Thanks to Reviewers for their suggestions in improving the manuscript. Authors have taken reviewers' comments and suggestions seriously. Authors have answered and have addressed all the concerns and questions mentioned by the reviewer. Professor Hsiang-te Kung (University of Memphis) has helped us improve the grammar and edit the manuscript carefully. Please see the revised version of the paper.

**Q:** In my opinion, maybe the "Discussions" as an independent part is better. Also, the authors should improve the Discussions in the section "conclusions and discussions" The subject is interesting and meaningful. I think it can be published after minor revision.

**Reply:** The authors have revised in the section "conclusions and discussions" based on Revivers suggestion, the "Discussions" as an independent part, and it's impovered by authors. Modified parts have been marked in red in the revised paper. Please see the article on page 9-10, lines to 339-389 in revised manuscript with obviously marked. Please see as follows:

**3 Discussion**

Given the seasonal differences, the unbalanced distribution of precipitation resulted in an apparent variation in the surface runoff and further imbalanced the spatial distribution of the water quality in the research zone (Fan et al.,2012; Prathumratana et al.,2008; Li et al.,2015). During the wet season (May) in the Jinghe Oasis, melted water from mountain ice and snow promote the flow in the Jing River, thereby resulting in a significant increase in surface runoff leading to an improved water quality in the rainy season compared to the dry season. During the

dry season, the aquatic plants in rivers and lakes grow as the temperature rises, which can, to a certain degree, absorb and purify part of the water quality parameters. Therefore, significant and seasonal changes in the surface runoff at the research zone are important factors resulting in noticeable differences in the spatial distribution of water quality characteristics during the wet and dry seasons. Another major factor that causes differences in the spatial distribution of water quality is the change in land use/cover, especially farmland. During the dry season, farmland areas have a greater influence on more water quality variables than they do during the rainy season because of their intensive fertilization and agricultural runoff from soil erosion (Ngoye et al, 2004;Li et al., 2009; Tran et al., 2010). Multiple factors threaten the ecological safety of the Jinghe Oasis system. Especially in recent years, the lakeside desertification zone has rapidly expanded because of the decrease in the Ebinur Lake area and the degradation of lakeside vegetation under influence of the strong winds in Alashankou. In the current overall situation, the human activities influencing land use/cover changes are directly related to the development of the vulnerable ecological area that surrounds Ebinur Lake.

Recent statistics indicate that the annual growth rate of the population in the Jinghe Oasis is approximately 2.49%, which is slightly higher than previous growth rates (Li, 2006). Under the stress of a large population, the amount of inappropriate activities that negatively impact land use/cover in the Jinghe Oasis will increase. For the last 30 years, cotton has been the major crop in the Jinghe Oasis. The results of the current study indicate that the sampling points surrounding the farmland in the research zone have lower water quality values than the other studied regions. The primary livelihoods of the urban residents around the Ebinur Lake area are agricultural and animal husbandry industries. Pollutants that result in high TP and $NH_3-N$ contents in water include the excessive application of chemical fertilizers on farmlands, the production of livestock manure in rural villages, randomly stocked garbage, and domestic wastewater. The improper application of chemical fertilizers and pesticides to a vast area leads to high water nitrogen and phosphorus contents, resulting in the spread of algae in river sections. Consequently, the amount of dissolved oxygen in the river may decrease, the water quality may deteriorate, and eutrophication may occur. Furthermore, this scenario poses a serious salinization problem. Certain measures have been implemented for the ecological protection of Ebinur Lake, such as returning farmland to forest, cultivating ecological forests, and promoting efficient irrigation and water-saving technologies. However, these measures promote the gradual expansion of the lake area and result in different degrees of negative consequences. The most apparent result has been the rise of the underground water level, which has aggravated land salinization in the lowland areas and resulted in vast expanses of uncultivated former agricultural lands. Statistics indicate that soil salinization in the Ebinur Lake area mainly occurs in Bortala River, Jing River, the villages and towns surrounding Ebinur Lake, areas downstream of the Daheyanzi River, and areas north of Bole City (Mi et al.,2010). Severe soil

salinization has seriously affected the farming of crops; therefore, some farmers have increased the amount of chemical fertilizers they apply to increase yield, which also increases the pollution of the water and soil. Others have even abandoned their land, thereby causing land use/cover change.

Most rivers in Xinjiang are characterized by a low water yield, short flow, small water environmental capacity, poor self-cleaning capability, and low tolerance to pollution. Hence, an artificial change in the land use and exploration of resources in lake regions lead to an evident correlation between land use/cover types and water quality. In addition, the scientific utilization and protection of the water resources of Ebinur Lake and the scientific application of chemical fertilizers and improvement of their application rates are important actions and should be addressed to achieve sustainable development in the agricultural irrigation zones of the Jinghe Oasis and rivers of Xinjiang.

In addition, authors have revised the figures and tables as well as words expression in the entire manuscript, please see the text.

Authors tried our best to improve the manuscript and made some changes in the manuscript. Authors appreciate for Editors/Reviewers' warm work earnestly, and hope that the correction will meet with approval. Once again, thank you very much for your comments and suggestions.

All in all, if you have any questions about our paper, please contact with me as follow address:

E-mail:zhangfei3s@163.com

Thanks very much.

Best wishes and warmly regards for you.

Sincerely yours Fei ZHANG

11[th], Jan., 2017

---

## Author Comment (AC3) · 11 Jan 2018

**Replies to the comments and suggestions**

**Dear editor and Anonymous Referee #2:**

Thank you for your letter and for the reviewers' comments concerning our manuscript entitled" Recognition of spatial framework for water quality and its relation with land use/cover types from a new perspective: A case study of Jinghe Oasis in Xinjiang, China". Those comments are all valuable and very helpful for revising and improving our paper, as well as important guiding sense to our researches. The authors have studied comments carefully and have made correction which we hope meet with approval. Revised portions are marked in red in the paper and authors have tried our best to revise the manuscript based on their suggestions. The point-by-point responses to each of the comments are presented as follows.

**Anonymous Referee #2**

This manuscript conducted a very detailed work on investigating the water quality and its relation with land use/cover types. While the data and samples have been processed carefully and rigorously, the results look reasonable and trustful. However, I have several major concerns regarding to this work.

**Reply:** Thank you for your positive evaluation and encouragement. We have carefully revised the paper according to your comments and suggestions. The responses to the specific comments are presented below

**Q:** There are two key words in the title that I cannot find strong evidence from the context to support, the "spatial" and "a new perspective". I cannot see how the authors make their findings 'spatial', since there is nothing that has been shown as a map, except for the study area (Figure 1) and land cover (Figure 6). From the Introduction, I also cannot see why the authors labeled their work as "a new perspective"? It is new because of what? This should be made clear in the Introduction.

**Reply**: Thanks to Reviewers for their suggestions in improving the manuscript, based on reviewer 1, the title is revised; 'spatial' is changed "spatial pattern". "a new perspective" is deleted by authors.

The title has changed into "Recognizing spatial patterns in water quality and their relation with land use/cover types: A case study of the Jinghe Oasis in Xinjiang, China"

**Q:** I am also unclear about the role that remote sensing data played in this study. What was it for, only for revealing the land use/cover types of each sample? How was it used? It was said there was a 1km buffer zone established for each sample. Then how? And why 1km? Besides, in line 246, there is the accuracy of land use/cover classification. I am wondering how these accuracy numbers were derived? Using what as reference

(ground truth)?

**Reply**: In order to extract the area of land use/cover types, therefore, remote sensing data is selected in this study.

1km buffer zone established for each sample in this study, based on most research (Wang et al., 2016; wang et al., 2017) and regional natural situation, Wang et al., (2016) reports a correlation between croplands areas and water quality in rivers, but this correlation isn't significant in 500 m. Wang et al., (2017) reports a significant correlation between land use and cover areas and water quality in kilometer scale in rivers. Because the study area is located in the arid areas of Xinjiang, desert dust and salt dust are major environmental hazards in this study area. Desert dust and salt dust seriously affect the atmosphere and water quality and accelerate the degradation of vegetation and threaten ecological security in the oasis. However, with economic development of and population growth, human improvement of desert land and saline land increases the area of croplands (Yu et al., 2017), and small changed in landscape. Therefore, 1km buffer zone established for each sample in this study.

The accuracy of land use/cover classification is required by confusion matrix, historical data, high-resolution Google Earth images, and field survey data, selected to verify that more than 100 pixels of each land cover type were used for the training data, and the confusion matrix to verify the classification results. From the classification results obtained in May and October 2015 (Figure 6), precision has increased to 89.9750% and 86.2848%, and the kappa coefficients are 0.8681and 0.8184, respectively by confusion matrix (Table 2). Please see the article on page 6-7, lines to246-250 in revised manuscript with obviously marked as follows:

In this study, historical data, high-resolution Google Earth images, and field survey data were selected to verify that more than 100 pixels of each land cover types were used for the training data, and the confusion matrix was used to verify the classification results. For the classification results obtained in May and October 2015 (Figure 6), precision increased to 89.9750% and 86.2848%, respectively, and the kappa coefficients were 0.8681 and 0.8184, respectively, based on the confusion matrix (Table 2),

Table 2 The calculation of a confusion matrix by a maximum likelihood supervised classification

|  | LULC | Water body | Saline land | Farmland | Forest grassland | Other land | Total | User's accuracy (%) |
|---|---|---|---|---|---|---|---|---|
|  | Water body | 144 | 0 | 0 | 0 | 0 | 144 | 100 |
|  | Saline land | 0 | 77 | 0 | 0 | 16 | 93 | 82.79 |
|  | Farmland | 0 | 36 | 101 | 0 | 0 | 137 | 73.72 |
| May | Forest-Grass land | 0 | 36 | 0 | 101 | 0 | 137 | 73.72 |
|  | Other land types | 1 | 0 | 0 | 0 | 87 | 88 | 98.96 |
|  | Total | 145 | 149 | 101 | 101 | 103 | Overall=89.9750% | |
|  | Producer's accuracy (%) | 99.31 | 51.67 | 100 | 100 | 84.46 | Kappa=0.8681 | |

| | | | | | | | | |
|---|---|---|---|---|---|---|---|---|
| | Water body | 144 | 0 | 0 | 0 | 0 | 144 | 100 |
| | Saline land | 0 | 57 | 0 | 0 | 26 | 83 | 86.67 |
| | Farmland | 0 | 16 | 101 | 0 | 0 | 117 | 86.32 |
| October | Forest-Grass land | 4 | 16 | 0 | 101 | 0 | 117 | 86.32 |
| | Other land types | 0 | 0 | 0 | 0 | 77 | 77 | 100 |
| | Total | 148 | 89 | 101 | 101 | 103 | Overall=86.2848% | |
| | Producer's accuracy (%) | 97.29 | 64 | 100 | 100 | 74.75 | Kappa=0.8184 | |

**Reference**

Wang XP, Zhang F, Li XH, Cao C, Guo M. 2017. Correlation analysis between the spatial characteristics of land use/cover-landscape pattern and surface-water quality in the Ebinur Lake area. Acta Ecologica sinica, 2017, 37(22):7438~7452

Wang J, Zhang F, Zhang Y, Ren Y, Yu HY, 2016. Correlation between the spatial water quality and land use/cover in the Ebinur Lake area Acta Ecologica sinica, 2016,36(24):7971~7980.

Yu, H., Zhang, F., Kung, H. T., Johnson, V. C., Bane, C. S., & Wang, J., et al. 2017. Analysis of land cover and landscape change patterns in ebinur lake wetland national nature reserve, china from 1972 to 2013. Wetlands Ecology & Management(3), 1-19.

**Q:** The grammar and usage have pervasive problems. It needs to be comprehensively edited by someone with strong English language skills before publication. There are some words, such as 'framework' in the title, and 'layers' in line 19, that are very confused to me. I have no idea what they are referring to.

**Reply**: Thanks to Reviewers for their suggestions in improving the manuscript, based on reviver 1, the title is revised, 'spatial framework' is changed ""spatial pattern". But then Professor **Hsiang-te Kung (University of Memphis)** has helped us improve the grammar and edit the manuscript carefully. Please see the revised version of the paper.

**Q:** Line 132, Davies-Bouldin index just comes out suddenly. What is this? What is the reference for it? What do you mean by "Through the K-means :"? These content needs to be rewritten.

**Reply**: The Davies-Bouldin clustering index was used to determine the optimal number of the clusters for a dataset. The sections of about Davies-Bouldin clustering index is rewritten by authors. Please see the article on page 4, lines to 129-133 in revised manuscript with obviously marked. As follows:

The Davies-Bouldin clustering index was used to determine the optimal number of clusters for the dataset (An et al., 2016; Park et al., 2014). The lower the Davies–Bouldin index value is, the better the clusters are

differentiated. The K-means cluster analysis was combined with the Davies–Bouldin index (DBI) to select the clustering number. (Zhou et al., 2016).

**Reference**

An, Y., Zou, Z., & Li, R. 2016. Descriptive characteristics of surface water quality in hong kong by a self-organising map. International Journal of Environmental Research & Public Health, 13(1):115.

Park, Y. S., Kwon, Y. S., Hwang, S. J., & Park, S. 2014. Characterizing effects of landscape and morphometric factors on water quality of reservoirs using a self-organizing map. Environmental Modelling & Software, 55(5):214-221.

**Q:** What is the DCA gradient axis in line 163? Why do you say "Therefore, the redundancy analysis (RDA) method was applied to". I cannot see connections between the sentences before the word 'Therefore' and those after it.

**Reply**: The section is revised by authors. Please see the article on page 4, lines to 161-165 in revised manuscript with obviously marked. As follows:

The results showed that the DCA gradient shaft length was less than 3. Based on the results of Wang et al (2017), when the DCA gradient shaft length is less than 3, the redundancy analysis (RDA) method can explore the relationship; therefore, the redundancy analysis (RDA) method was applied to determine the influence trend of land use/cover changes on the water quality within the Ebinur Lake buffer area.

**Reference**

Wang XP, Zhang F, Li XH, Cao C, Guo M. 2017. Correlation analysis between the spatial characteristics of land use/cover-landscape pattern and surface-water quality in the Ebinur Lake area. Acta Ecologica sinica 2017,37(22):7438~7452

**Q:** Sentence in line184-185 needs to be rewritten.

**Reply**: Thanks to Reviewers for their suggestions in improving the manuscript, the sentence is rewritten, Please see the article on page 5, lines to 183-184 in revised manuscript with obviously marked. As follows:

Six clusters were formed according to the DBI, where minimal value is at six clusters as Figure 3a.

In addition, authors have revised the figures and tables as well as words expression in the entire manuscript, please see the text.

Authors tried our best to improve the manuscript and made some changes in the manuscript. Authors appreciate for Editors/Reviewers' warm work earnestly, and hope that the correction will meet with approval. Once again, thank you very much for your comments and suggestions.

All in all, if you have any questions about our paper, please contact with me as follow address:

E-mail:zhangfei3s@163.com

Thanks very much.

Best wishes and warmly regards for you.

Sincerely yours Fei ZHANG

11th, Jan., 2017

---

## Author Comment (AC4) · 11 Jan 2018

**Replies to the comments and suggestions**

**Dear editor and C. Huang:**

Thank you for your letter and for the reviewers' comments concerning our manuscript entitled" Recognition of spatial framework for water quality and its relation with land use/cover types from a new perspective: A case study of Jinghe Oasis in Xinjiang, China". Those comments are all valuable and very helpful for revising and improving our paper, as well as important guiding sense to our researches. The authors have studied comments carefully and have made correction which we hope meet with approval. Revised portions are marked in red in the paper and authors have tried our best to revise the manuscript based on their suggestions. The point-by-point responses to each of the comments are presented as follows.

1. "Spatial framework" in the title and in the content sounds a little bit weird, maybe replaced with spatial pattern or spatial distribution?

**Reply:** Thanks to Revivers for encouraging Authors to revise the manuscript. Thanks to Reviewers for their suggestions in improving the manuscript. Authors have taken reviewers' comments and suggestions seriously. The title is revised, "Spatial framework" change into "spatial pattern". Modified parts have been marked in red in the revised paper.

**2. Line 132, repeated sentence.**

Reply: Line 132, repeated sentence has deleted by the authors.

3. Fig. 3, it seems there is only one sample in cluster 3? How much significance would the results of this cluster have? Should this cluster be deleted and keep only 5 clusters?

**Reply:** Only one sample of cluster 3 cannot be deleted, because the results are automatic output by SOM clustering. The results show the water quality sample is special of water quality. For significance of the cluster results, the SOM technique is a powerful tool to group the similar input patterns from a multidimensional input space into a much lower dimensional space, usually two dimensions; SOM can be used for clustering, classification, estimation, prediction, and data mining (Yan et al., 2016; Park et al., 2014), the spatial distribution of water quality is classified by SOM technique. In the present study, the main objectives of this study were analysis the relationship between water quality parameters and land use/cover types in different stages. The water samples are divided into six categories, which references the results of previous studies and consider of the status

of water quality (Park et al., 2014; Ren et al., 2017).

**Reference**

Yan, A., Zou, Z., & Li, R. 2016. Descriptive characteristics of surface water quality in hong kong by a self-organising map. International Journal of Environmental Research & Public Health, 13(1):115.

Park, Y. S., Kwon, Y. S., Hwang, S. J., & Park, S. 2014. Characterizing effects of landscape and morphometric factors on water quality of reservoirs using a self-organizing map. Environmental Modelling & Software, 55(5):214-221.

Ren Y, Zhang F, Wang J, Zhang Y, Yang ST. 2017. Spatio-temporal characteristics and source identification of surface water pollutants in Lake Ebinur Watershed, Xinjiang. Journal of Lake Sciences, 2017, 29(5):1143-1157.

4. Table 1, it seems there is no difference between cluster 1 and cluster 2? Then, what makes them two clusters?

**Reply:** Thanks to Reviewers for their suggestions in improving the manuscript. Table 1 shows the results of water quality combined with Chinese Environmental Quality Standard for Surface Water (GB 3838-2002), clusters 1 to 6 shows different water quality standard. Clusters 1 and 2 have identical water quality classification level, and their COD and TN contents are higher than the standard values. The result of this classification is not a fixed value, but an interval value. the SOM technique is a powerful tool to group the similar input patterns from a multidimensional, usually two dimensions; SOM can be used for clustering, classification, estimation, prediction, and data mining (Yan et al., 2016; Park et al., 2014), the methods can mining more detailed spatial distribution patterns, there is difference between cluster 1 and cluster 2 in more detailed spatial distribution patterns.

**Reference**

Yan, A., Zou, Z., & Li, R. 2016. Descriptive characteristics of surface water quality in hong kong by a self-organising map. International Journal of Environmental Research & Public Health, 13(1):115.

Park, Y. S., Kwon, Y. S., Hwang, S. J., & Park, S. 2014. Characterizing effects of landscape and morphometric factors on water quality of reservoirs using a self-organizing map. Environmental Modelling & Software, 55(5):214-221.

5. Table 2, it would be better to add lines to separate each clusters.

**Reply:** Thanks to Reviewers for their suggestions in improving the manuscript. According to Reviewers for their suggestions, lines are added in the Table 2. Modified parts have been marked in red in the revised paper. Please see the article.

6. Line 239, "combined with" should be "according to".

Reply: Thanks to Reviewers for their suggestions in improving the manuscript. "Combined with" changed into

"according to". Modified parts have been marked in red in the revised paper. Please see the article on page 6, lines

to 239 in revised manuscript with obviously marked.

7. Line 245-246, how was the accuracy derived for land cover maps in Figure 6?

**Reply:** The accuracy classification for land cover maps in Figure 6 is required by Land cover transition matrix. Confusion matrix is added by the authors, Please see the article, as follows:

Table 2 The calculation of a confusion matrix by a maximum likelihood supervised classification

|         | LULC                    | Water body | Saline land | Farmland | Forest
grassland | Other
land | Total            | User's   |
|---------|-------------------------|------------|-------------|----------|---------------------|---------------|------------------|----------|
|         |                         |            |             |          |                     |               |                  | accuracy |
|         |                         |            |             |          |                     |               |                  | (%)      |
|         | Water body              | 144        | 0           | 0        | 0                   | 0             | 144              | 100      |
|         | Saline land             | 0          | 77          | 0        | 0                   | 16            | 93               | 82.79    |
|         | Farmland                | 0          | 36          | 101      | 0                   | 0             | 137              | 73.72    |
| May     | Forest-Grass land       | 0          | 36          | 0        | 101                 | 0             | 137              | 73.72    |
|         | Other land types        | 1          | 0           | 0        | 0                   | 87            | 88               | 98.96    |
|         | Total                   | 145        | 149         | 101      | 101                 | 103           | Overall=89.9750% |          |
|         | Producer's accuracy (%) | 99.31      | 51.67       | 100      | 100                 | 84.46         | Kappa=0.8681     |          |
|         | Water body              | 144        | 0           | 0        | 0                   | 0             | 144              | 100      |
|         | Saline land             | 0          | 57          | 0        | 0                   | 26            | 83               | 86.67    |
|         | Farmland                | 0          | 16          | 101      | 0                   | 0             | 117              | 86.32    |
| October | Forest-Grass land       | 4          | 16          | 0        | 101                 | 0             | 117              | 86.32    |
|         | Other land types        | 0          | 0           | 0        | 0                   | 77            | 77               | 100      |
|         | Total                   | 148        | 89          | 101      | 101                 | 103           | Overall=8        | 6.2848%  |
|         | Producer's accuracy (%) | 97.29      | 64          | 100      | 100                 | 74.75         | Kappa=0.8        | 3184     |

In addition, authors have revised the figures and tables as well as words expression in the entire manuscript, please see the text.

Authors tried our best to improve the manuscript and made some changes in the manuscript. Authors appreciate for Editors/Reviewers' warm work earnestly, and hope that the correction will meet with approval. Once again, thank you very much for your comments and suggestions.

All in all, if you have any questions about our paper, please contact with me as follow address:

E-mail:zhangfei3s@163.com

Thanks very much.

Best wishes and warmly regards for you.

Sincerely yours Fei ZHANG 11th, Jan., 2017

**1 Recognizing spatial patterns in water quality and their relation with**

**2 land use/cover types: A case study of the Jinghe Oasis in Xinjiang,**

**3 China**

**4 Fei ZHANG1,2,3\* Juan WANG4 Xiaoping WANG1,2,3**

- 5 1. College of Resources and Environment Science, Xinjiang University, Urumqi, Xinjiang 830046
- 6 2. Key Laboratory of Oasis Ecology, Xinjiang University, Urumqi, Xinjiang 830046
- 7 3. Key Laboratory of Xinjiang wisdom city and environment modeling, Urumqi, Xinjiang 830046
- 8 4. College of Geography and Remote Sensing Science, Beijing Normal University, 100875, Beijing
- 9 Abstract: To understand the relation between spatial water quality patterns and changes in land use/cover types in the Jinghe
- Oasis, this study divided 47 water sampling sites, which were measured in May and October 2015, into 6 cluster layers using
   the self-organizing map (SOM) method based on non-hierarchical *k*-means classification. Next, it determined the distribution
- 12 characteristics of water quality indices for the time sequence. The water quality indices included the chemical oxygen
- 13 demand (COD), biological oxygen demand (BOD), suspended solids (SS), total phosphorus (TP), total nitrogen (TN),
- 14 ammonia nitrogen (NH3-N), chromaticity (SD), and turbidity (NUT). Based on the results, we collected data on the changes
- 15 in the farmland, forest-grass land, water body, salinized land, and other land types during the wet and dry seasons and
- 16 combined these data with the classification results of the GF-1 remote sensing satellite data obtained in May and October
- 17 2015. We then discussed the influences of land use/cover type on water quality for different layers and seasons. The results
- 18 indicate that Clusters 1 to 3 included monitoring samples from the wet season (May 2015), whereas Clusters 4 to 6 included
- 19 monitoring samples from the dry season (October 2015). In general, the COD, SS, NUT, TN, and NH3–N values were high
- 20 in Clusters 1 and 2. The SD values for Clusters 1, 4, and 6 were high. Moreover, high BOD and TP values were mainly
- 21 concentrated in Clusters 4 and 6. In the discussion on the relation between the different water quality parameters and land
- 22 use/cover type changes, we determined that farmland, forest-grassland, and salinized land significantly influenced the water
- 23 quality parameters in the Jinghe Oasis. In Clusters 1, 2, and 6, the size of the water area, to a certain extent, also influenced
- 24 changes in the water quality parameters. In addition, the influences of various land use/cover types on the water quality
- 25 parameters in the research zone during the different seasons exhibited the following descending order of magnitude:
- 26 farmland  $\rightarrow$  forest-grass land  $\rightarrow$  salinized land  $\rightarrow$  water body  $\rightarrow$  others. Moreover, their influences were lower during the
- 27 wet season than the dry season. In conclusion, developing research on the relation between the spatial framework of the
- 28 water quality in the Jinghe Oasis and land use/cover type changes is significant for the time sequence distribution of water
- 29 quality in arid regions from both theoretical and practical perspectives.
- 30 Key words: SOM; Water quality spatial distribution; Land use/cover; Correlation analysis; GIS

**31 **0 Introduction**

Water quality is of great importance to the study of water resources in arid regions. Accurate information on the spatial distribution of surface water quality is imperative for assessing environmental monitoring, land-surface water management and watershed changes (NRC, 2008; Sun et al., 2012). Land use/cover changes in drainage basins significantly influence the water quality of rivers, lakes, river mouths, and coastal areas (Huang et al., 2013a; Bu et al., 2014; Hur et al., 2014). Surface water resources, through runoff or infiltration, will always carry a large amount of pollutants (Swaney et al., 2012). Therefore, the spatial allocation of land use and land cover changes in drainage basins frequently influences or even

\* Corresponding author: Zhangfei Tel: +86 13579925126 E-mail address: zhangfei3s@163.com

39 endangers water quality through non-point source pollution (Swaney et al., 2012). However, the regional 40 differences and complexity of land use/cover types result in various relations between land use/cover and 41 water quality in different regions (Yang et al., 2016). Therefore, it is very important to explore the relationship between land use/cover types and water quality for the development and management of the 42 43 basin. (Uuemaa et al., 2005; Xiao et al., 2007; Wan et al., 2014). At present, numerous scholars have 44 extensively applied statistical methods to determine the mutual relations between land use/cover changes 45 and water quality in various research zones (Céréghino et al., 2009; Bierman et al., 2011; Huang et al.,2013b). These methods include correlation analysis (Lee et al., 2009; Li et al., 2015), multiple 46 47 regression (Park et al., 2014), and redundancy analysis (De et al., 2008; Shen et al., 2015).

48 A self-organizing map (SOM) is a type of artificial neural network algorithm; it is a self-organizing 49 and self-learning network visual method that can express multi-dimensional spatial data in 50 low-dimensional points through non-linear mapping (Kohonen., 2001). A SOM is an all-purpose 51 classification tool that can connect samples with variables (Kohonen., 2013; Zhou et al., 2016). In recent 52 years, SOMs have become increasingly popular in environmental research because of their capacity to 53 address non-linear relations. Kalteh (2008) and Céréghino (2009) discussed the application of the SOM 54 method in environmental science, particularly in water resource classification. Chon (2011) evaluated the 55 application of SOM technology in the field of ecology. The high-dimensional, non-linear, and uncertain 56 features of water quality monitoring data result in a certain complexity during the analysis and evaluation 57 of surface water quality data. Therefore, data mining and the modern mode recognition method have been 58 introduced to analyse and explain water quality monitoring data, which can, to a certain extent, offset the 59 deficiency of the traditional method (Li et al., 2013). The Jinghe Oasis, which comprises an oasis and a 60 desert, is a typical mountainous zone in an arid region and an important part of the northern slope of the 61 Tianshan Mountains. Under the influences of the drainage basin climate and human activities, the pollution 62 of the regional ecological environment, which results from agricultural and domestic wastewater sources 63 around the Jinghe Oasis that are directly discharged or discharged through the river, has become an urgent 64 problem related to sustainable socioeconomic development in Xinjiang. Therefore, a typical section of the 65 Jinghe Oasis in the plain area of the arid region was selected as the research object of this study. The SOM method was applied to recognize the spatial distribution of water quality in the Jinghe Oasis. Based on the 66 67 result, this study offers a tentative exploration of the relation between water quality and land use/cover 68 changes in different clusters and provides new insights on controlling, managing, and protecting the 69 ecological environment in the Jinghe Oasis.

The main objectives of this study were to (1) analyse the spatial framework of water quality using the self-organizing map (SOM) method based on non-hierarchical k-means classification; (2) explore the relationship between the water quality parameters and land use/cover types in different clusters; and (3) analyse the relationship between water quality parameters and land use/cover types in different stages.

**74 **1 Materials and methods**

**75 **1.1 Study area**

The Jinghe Oasis is in the centre of Eurasia in the northwest Xinjiang Uygur Autonomous Region at 44°02'~45°10'N and 81°46'~83°51'E. The Jinghe Oasis is composed of wetland and desert oasis vegetation and wildlife and is a national desert ecological reserve. The study area has a unique wetland ecological environment, and it has been listed as the Xinjiang Uygur autonomous region "wetland nature reserve". The region has 385 kinds of desert plants, approximately 64% of the vast amount of desert plants 81 in China. The Jinghe Oasis was once fed by 12 river branches belonging to three major river systems, the 82 Bortala River, Jing River and Kuytun River, which were mainly rivers connected with Ebinur Lake. Due to 83 natural environmental changes and human activities (i.e., modern oasis agricultural development), many 84 rivers gradually lost their hydraulic connections with Ebinur Lake, and only Bortala River and Jing River 85 currently supply water to the lake. The climate in the Jinghe Oasis is a typical continental arid climate, with an annual average temperature of 7.36 °C, an average precipitation of 100~200 mm, and an average 86 87 evaporation of 1500~2000 mm (Zhang et al., 2015). In recent years, under the dual influence of natural and 88 human factors, the water resources of the Jinghe Oasis have degraded seriously, causing an extreme 89 decrease in the natural oasis and water area, desertification of the land, salinization of the farmland, serious 90 grassland degradation, and water quality salinization (Jilil et al., 2002). At the same time, under the effect of 91 the strong winds in Alashankou, the region has become a main source of dust; this affects the ecological 92 environment of northern Xinjiang. This study, by using actual data mining, established the research area 93 shown in Figure 1.

94 95

Fig. 1 Location of the study area

96

97 **1.2 Data acquisition and processing**

98 (1) We applied GF-1 remote sensing images obtained in May and October 2015 as the data sources 99 (see http://www.cresda.com/CN/). These images were not influenced by clouds, fog, or snow cover, and 100 their quality was good. We conducted radiation and orthographic corrections for the remote sensing image 101 data combined with 1:50,000 scale digital elevation model (DEM) data. We established five land use/cover 102 types by using the Environment for Visualizing Images software (ENVI Version 5.0), namely, farmland, 103 forest-grassland, water body, salinized land, and others, based on the actual conditions of the research zone. 104 Finally, we generated a vector data map of the land use/cover types for two stages of the research zones.

105 (2) The pillar industries in the Jinghe Oasis include salt production and Artemia breeding. No heavy 106 industry is present; thus, point source pollution from industrial wastewater was not considered in the 107 research zone. Research samples were collected from agricultural land in Jinghe County and Tuotuo 108 Village, which surround Ebinur Lake, a national ecological zone in Ebinur Lake Bird Isle, and the Ganjia 109 Lake Haloxylon natural conservation area. In total, we collected 47 water samples, with 23 collected in 110 May and 24 collected in October 2015. The monitoring indices used included chemical oxygen demand 111 (COD), five-day biological oxygen demand (BOD5), suspended solids (SS), total phosphorus (TP), total 112 nitrogen (TN), ammonia nitrogen (NH3-N), chromaticity (SD), and turbidity (NUT). All polyethylene 113 bottles were used to store the samples. The bottles were cleaned, dried, and sealed with deionized water 114 before sampling. The samples were taken to the laboratory for the measurements and analyses after 115 collection. We applied dichromate titration, dilution and inoculation, gravimetry, ammonium molybdate 116 spectrophotometry, alkaline potassium persulfate decomposition UV spectrophotometry, and Nessler 117 reagent spectrophotometry to measure the COD, BOD5, SS, TP, TN, and NH3-N, respectively. The 118 analyses of all the samples were entrusted to and completed by Urumqi Jincheng Measurement Technology 119 Co., Ltd.

**1.3 Recognition of water quality spatial characteristics based on the SOM method with non-hierarchical *k*-means classification**

122 At present, classifications based on the SOM neural network are mainly unsupervised and applied to

123 fault analyses, text clustering, and water quality evaluations. The method, which does not require a 124 consistent data distribution, is simple and can address detailed information without the influence of minor 125 local problems. The results are the distribution features of the input mode and topological structures (Li et 126 al.,2010). A typical SOM network generally consists of input and output clusters. All the nerve cells in the 127 input cluster and the weight vectors in the output cluster are connected and classified as typed data using 128 the SOM via a learning process. Accordingly, the k-means algorithm is applied to keep each cluster 129 compact and to separate the clusters from each other as much as possible. The Davies-Bouldin clustering 130 index was used to determine the optimal number of clusters for the dataset (An et al., 2016; Park et al., 131 2014). The lower the Davies–Bouldin index value is, the better the clusters are differentiated. The K-means 132 cluster analysis was combined with the Davies-Bouldin index (DBI) to select the clustering number. (Zhou et al., 2016). 133

134 The SOM method based on non-hierarchical k-means classification was applied to the spatial 135 framework of the water quality in the research zone by implementing the following steps. (1) We input the water sample data for clustering from May and October 2015 to the SOM network. We applied the 136 topological values for calculating the network size to select the quantity of nerve cells and determine the 137 output results based on the minimum values of the Quantization Error (QE) and Topological Error (TE). 138 139 The OE was used to determine the capacity of the established neural network in distinguishing the original 140 input data, whereas the TE was used to measure the neural network quality, i.e., to evaluate whether the 141 network was applicable for training (Kohonen, 2001). After determining the network size, we conducted network training and obtained a set of weight values. (2) The weight value obtained from the SOM 142 143 clustering results was considered the initial cluster centre, and the k-means algorithm was initialized to execute this algorithm and, combined with the DBI index, select the clustering number. This clustering 144 145 combination algorithm maintains the self-organizing features of the SOM network, inherits the high 146 efficiency of the k-means algorithm, and offsets the poor clustering effects that result from the excessive 147 convergence time of the SOM network and the inappropriate selection of the initial clustering centre for 148 the k-means algorithm. The SOM requires a SOM toolbox and some basic functions in Matrix Laboratory 149 (MATLAB) (Zhang, 2015). This study used MATLAB 2013a as the calculation platform.

**150 **1.4 Spatial analysis of the influences of land use/cover change on water quality**

151 As an artificial system disturbance, land use/cover type is the second major boundary condition that 152 directly or indirectly influences hydrologic processes and exerts a considerable effect on the drainage water environment. First, we obtained information on the land use/cover types within the 1 km buffer zone of the 153 water quality sampling points in the research area using the spatial analysis function of ArcGIS 9.3. Based 154 155 on the results, we then discussed and analysed the correlation between water quality and land use/cover type changes at different levels and periods. For different levels, we established the correlation between 156 157 water quality and land use/cover types in each layer and discussed the influence of land use/cover type 158 changes on water quality. For different periods, analysed the land use/cover type information and eight 159 types of water quality indices during the dry and wet seasons. The land use/cover type information and 160 eight water quality indices were imported into Canoco 4.5 (Ter Braak and Smilauer, 2002) to test the DCA gradient axis. The results showed that the DCA gradient shaft length was less than 3. Based on the results 161 of Wang et al (2017), when the DCA gradient shaft length is less than 3, the redundancy analysis (RDA) 162 163 method can explore the relationship; therefore, the redundancy analysis (RDA) method was applied to 164 determine the influence trend of land use/cover changes on the water quality within the Ebinur Lake buffer 165 area. This method indicates the contribution rate of a single land use/cover variable on water quality and

- can directly demonstrate the correlation between land use/cover type and water quality parameters via a 2D
   ordination graph. The methodology is explained in the following section, and a conceptual flow chart
   describing the methodology is shown in Figure 2.
- 169
- 170

Fig. 2 Conceptual model of the methodology

**171 **2 Results and analysis**

**172 **2.1 Spatial framework of water quality in the Jinghe Oasis**

173 Regarding the network structure selection, neural networks with a more complicated structure 174 generally have a better capability to address complicated non-linear problems but require a longer training 175 time (Kohonen, 2013). Increasing the number of water quality indices can provide more abundant 176 information; however, the correlations among indices will also increase. The topological values were 177 selected to determine the grid size in this study, and the *k*-means clustering method was adopted to obtain 178 the results. Overall, after the standard processing of the water quality data, the best network training effect 179 was obtained from 35 (7×5) nerve cells, and the QE and TE values were 1.033 and 0.001, respectively.

When the average variance values are less than 5% for different clusters, the DBI is low; thus, the corresponding clustering number can be regarded as the best clustering result. Therefore, this study input the trained weights of the neuron nodes through the K-means cluster analysis combined with the DBI to select the clustering number. The results are shown in Figure 3a. Six clusters were formed because this number yielded the minimum DBI value (Figure 3a).

- 185
- 186 187

**Fig. 3** (a) Davies–Bouldin index plot. (b) Results of the SOM clustering of the cells on the map plane (distribution of the sampling sites on the SOM according to the eight water quality parameters and clustering of the trained SOM).

188

189 Figure 3b presents the results of the SOM clustering of the cells on the map plane, which exhibited 190 similarities among the different monitoring stations. In particular, Cluster 1 included the sampling points 191 around the Ganjia Lake Haloxylon natural conservation area in the southern Ebinur Lake region and points 192 east of Ebinur Lake and around the Kuitun River during the wet season. Cluster 2 included the monitoring 193 stations in the Jing River and around the agricultural ditch in the western Ebinur Lake region. Cluster 3 194 comprised samples of water from melted ice in the southern-western corner of the research zone, which 195 were grouped into only one type. Cluster 4 included the sampling points within the Ganjia Lake Haloxylon 196 natural conservation area during the dry season. Cluster 5 included the sampling points in the Jing River, 197 the agricultural ditch and around Ebinur Lake. Cluster 6 contained points located around the Kuitun River 198 and Ebinur Lake Bird Isle, which had more pools. In general, although individual points may interfere with 199 the explanation of the results, the classification results can identify the time sequence features in the 200 research zone. Clusters 1 to 3 entirely included samples from the wet season (May 2015), whereas Clusters 201 4 to 6 contained the monitoring samples from the dry season (October 2015). To further observe the 202 information on the water quality parameters of the Jinghe Oasis based on the response of different nerve cells, the water quality information from the various cluster groups was visualized. The results are shown 203 204 in Figure 4.

205

Fig. 4 The patterning results for the water quality parameters on the SOM plane

206

| 207 | Figure 4 shows the distribution of different water quality parameters on the SOM. High COD, TN,                          |
|-----|--------------------------------------------------------------------------------------------------------------------------|
| 208 | NH3-N, and SD values were recorded in the right corner of the SOM network, thereby indicating a                          |
| 209 | declining trend in the southern Ebinur Lake region and the surrounding Kuitun River. The values during                   |
| 210 | the wet season were higher than those during the dry season. High values of SS and NUT were observed in                  |
| 211 | the left corner of the SOM network, which falls within the scope of the agricultural ditch and the Jing                  |
| 212 | River areas during the dry season. This region is mainly distributed around the agricultural land area and is            |
| 213 | significantly influenced by human activities. High TP values were observed within the scope of the lower                 |
| 214 | left corner, which mainly focuses on the agricultural ditch and Jing River during the dry season. Crops                  |
| 215 | were mature during the dry season, and the farmland area was increased, which thereby increased the TP                   |
| 216 | value. In contrast, the distribution of the BOD 5 was different, indicating a declining trend from the centre |
| 217 | to the surrounding area. High BOD5 values were mainly observed downstream of the Ganjia Lake                             |
| 218 | Haloxylon natural conservation zone, which is surrounded by a salt field, thereby exerting a certain                     |
| 219 | influence on the surrounding water quality. The regional change in the water quality of the Jinghe Oasis                 |
| 220 | was reflected clearly and directly through the SOM. To observe the distribution of the water quality                     |
| 221 | parameters directly, we collected different values of the water quality parameters at various layers (Figure             |
| 222 | 5).                                                                                                                      |
|     |                                                                                                                          |

- 223
- 224

**Fig. 5 Average values for the water quality parameters**

225 Figure 5 shows that the distribution of water quality parameters varies in different clustering layers. Among the six clusters, water quality was generally relatively better in Cluster 3. However, Cluster 3 only 226 227 had one sampling point, which comprised ice and snow water. Therefore, its water quality was not 228 considered. In addition, the COD, SS, NUT, TN, and NH3-N contents were high in Clusters 1 and 2, which 229 indicates a relatively lower water quality compared with the water downstream of the Ganjia Lake 230 Haloxylon natural conservation zone, the surrounding Ebinur Lake region, and the agricultural land. The 231 SD values were high in Clusters 1, 4, and 6. Meanwhile, high BOD5 values were mainly concentrated in 232 Clusters 4 and 6, around the Ganjia Lake *Haloxylon* natural conservation zone and Ebinur Lake Bird Isle. Many pools in these clusters are, to a certain extent, influenced by the water area. The concentration 233 difference in the TP of each layer was minimal and considerably influenced by human activities, 234 particularly changes in the agricultural land area. Based on these results and Chinese Environmental 235 236 Quality Standards for Surface Water (GB3838-2002), we evaluated the grades of the water quality 237 parameters, including the COD, BOD5, TN, NH3–N, and TP, at different layers (Table 1).

238

**Table 1 The classes of water quality parameters in each cluster**

As shown in Table 1, according to Chinese Environmental Quality Standards for Surface Water (GB 3838-2002), clusters 1 to 6 did not satisfy the potable water quality level. Clusters 1 and 2 had an identical water quality classification level, and their COD and TN contents were higher than the standard values. In particular, the COD content exceeded level V. Meanwhile, the BOD5 and COD contents were excessive in Clusters 4 and 5, respectively. In Cluster 6, both the COD and BOD5 contents were excessive, but the COD content was higher.

**245 **2.2 Analysis of land use/cover type and its relation to water quality at different layers**

In this study, historical data, high-resolution Google Earth images, and field survey data were selected to verify that more than 100 pixels of each land cover type were used for the training data, and the confusion matrix was used to verify the classification results. For the classification results obtained in May

249 and October 2015 (Figure 6), precision increased to 89.9750% and 86.2848%, respectively, and the kappa 250 coefficients were 0.8681 and 0.8184, respectively, based on the confusion matrix (Table 2), which indicates an accurate classification result that satisfies the research requirements. Accordingly, ArcGIS was used to 251 252 establish a 1 km buffer zone around the water sampling points. The composition of land use/cover at 253 different layers was analysed according to the hierarchical results of the water quality parameters, and the 254 results are presented in Figure 7. 255 Fig. 6 The change in land use/cover of the Ebinur Lake area during the rainy (May) and dry (October) seasons in 2015 (a: 256 May; b: October) 257 Fig. 7 The area of land use/cover for each cluster 258 259 Table 2 The confusion matrix calculation with a maximum likelihood supervised classification 260 261 Figure 7 shows the land use/cover mode at different layers. In general, the salinization phenomenon 262 was serious across the entire research zone. Among the six clusters, Cluster 1 mainly included the sampling points around the Ganjia Lake Haloxylon natural conservation area, the eastern Ebinur Lake region, and 263 264 the Kuitun River. The major land type in this cluster was forest-grass land. The monitoring site in Cluster 2 265 was in the irrigation ditch of the Jinghe Oasis and in the Jing River. In May, crops do not grow abundantly 266 in the research zone, and the major land type in this cluster was forest-grass land. Cluster 4 mainly 267 included the Ganjia Lake Haloxylon natural conservation area and the sampling points in Tuotuo Village during the dry season, which include farmland and forest-grassland. The sampling points in Cluster 5 were 268 269 in the agricultural ditch, the Jing River, and the surrounding Ebinur Lake. The land types mainly included 270 forest-grassland and farmland, which was larger than the other land type. Cluster 6 was mainly located in 271 the Kuitun River and the surrounding Ebinur Lake Bird Isle. A considerable number of plants, such as reed, 272 grow in some of its pools. The percentages of the forest-grass land and salinized land within the 1 km 273 buffer were large, based on the actual conditions in the research zone and the distribution of the sampling 274 points. Based on these results, the correlation between land use/cover type and water quality parameters 275 from Clusters 1 to 6 was analysed. The results are presented in Table 3.

276

Table 3 The correlation coefficients between the land use/cover and water quality parameters in each cluster

277 In Cluster 1, the forest-grass land exhibited a negative correlation with the SS and NUT under the significance level of 0.01, with coefficients reaching up to -0.710 and -0.724, respectively. At a 278 279 significance level of 0.05, the water body type exhibited an obvious positive correlation with the COD and a negative correlation with the TN, with coefficients of 0.986 and -0.721, respectively. At a confidence 280 281 level of 0.01, the salinized land demonstrated a positive correlation with the NUT, with a coefficient of 282 0.756. In Cluster 2, the farmland presented a negative correlation with the COD and a positive correlation with the NUT at a confidence level of 0.05, and the coefficients were -0.581 and 0.639, respectively. 283 284 Under the same conditions, the forest-grass land exhibited a positive correlation with the COD, and the 285 coefficient was 0.613. At a confidence level of 0.01, the water body type exhibited an evident negative 286 correlation with the SS and NUT, and the correlation coefficients were -0.983 and -0.990, respectively. In 287 Cluster 4, several water quality parameters were mainly influenced by the farmland, forest-grass land, and salinized land types. At a confidence level of 0.05, the farmland exhibited a negative correlation with the 288 289 COD, with a coefficient of -0.652. At a confidence level of 0.01, the farmland demonstrated an evident 290 positive correlation with TP, and the coefficient is 0.872. At a confidence level of 0.05, salinized land 291 shows a clear negative correlation with TP and NUT, and the coefficients were -0.791 and -0.819, 292 respectively. In this layer, the others land type exhibited a positive correlation with the TP, with a

293 coefficient of 0.868. The sampling points in this layer were mainly located around Tuotuo Village, where 294 the influences of human activities are considerable; therefore, in Cluster 4, the correlation percentage of the others land type with TP was high. In Cluster 5, farmland demonstrated an evident negative correlation 295 296 with the BOD5 at a 0.01 confidence level, with a correlation coefficient reaching up to -0.881. At a 297 confidence level of 0.05, the salinized land showed a positive correlation with the BOD5, with a correlation 298 coefficient of 0.774. In Cluster 6, the forest-grass land showed a clear negative correlation at a confidence 299 level of 0.01, and the correlation coefficient reached -0.884. At a confidence level of 0.05, the water area 300 exhibited a positive correlation, with a correlation coefficient of 0.980.

301 From the results of the comprehensive analysis, the farmland, forest-grass land, and salinized land 302 types have a considerable influence on the water quality parameters in the Jinghe Oasis. In Clusters 1, 2, 303 and 6, the size of the water area, to a certain extent, also influenced changes in the water quality parameters. 304 Given the unbalanced distribution of the sampling points at different layers, the effect of the land use/cover 305 composition on the water quality in the research zone varied, so it can only indicate influences to a certain extent. Therefore, considering the actual conditions in Ebinur Lake, different land use/cover types and 306 307 water quality influences must be understood as a whole, and the correlation between the land use/cover types and water quality in the Jinghe Oasis at different periods is further discussed. 308

**309 2.3 Analysis of land use/cover changes in the Jinghe Oasis and their correlation with 310 water quality at different seasons**

The constituents of the land use/cover types at different seasons exerted diverse influences on the water quality. Therefore, an analysis of the influences of land use/cover type changes on water quality was conducted. The results are presented in Figure 8.

314

**Fig. 8 RDA analyses of comprehensive land use/cover and water quality (a: wet season; b: dry season)**

315 As shown in Figure 8, the farmland exhibited a negative correlation with the COD at a confidence 316 level of 0.01 during the wet season, with a correlation coefficient of -0.543. In contrast, it showed a 317 positive correlation with the NUT, with a correlation coefficient of 0.555. At a confidence level of 0.05, the farmland demonstrated a negative correlation with the NH3-N, and the correlation coefficient was -0.461. 318 319 At a confidence level of 0.05, the forest-grass land showed a positive correlation with the BOD5 and TP, 320 with correlation coefficients of 0.470 and 0.518, respectively. They exhibited a negative correlation with 321 the SS and NUT, with correlation coefficients of -0.529 and -0.498, respectively. At a confidence level of 322 0.05, the salinized land demonstrated a positive correlation with the BOD5 and TP, with correlation coefficients of -0.503 and 0.518, respectively. In contrast, it presented a negative correlation with the SS 323 and NUT, with correlation coefficients of 0.449 and 0.449, respectively. During the dry season, the 324 325 influence of the farmland on various water quality parameters evidently increased because of crop growth. At a confidence level of 0.01, the farmland presented a clear negative correlation with the COD and an 326 327 evident positive correlation with the TP, with correlation coefficients of -0.620 and 0.616, respectively. At 328 a confidence level of 0.05, the farmland showed a positive correlation with the TN and a negative 329 correlation with the BOD5, NH3-N and SD, with correlation coefficients of 0.543, -0.495, -0.522, and -0.526 for TN, BOD5, NH3-N and SD, respectively. At a confidence level of 0.01, the salinized land 330 demonstrated a negative correlation with the NUT and TP, and the correlation coefficients were -0.543 and 331 332 -0.603, respectively. In contrast, it presented a positive correlation with the BOD5 at a confidence level of 333 0.05 and a correlation coefficient of 0.522. Similarly, during the wet and dry seasons, the correlation 334 between the water body and others land types with the water quality parameters was small. Based on these

results, the influences of the various land use/cover types in the research zone on the water quality parameters exhibited the following order, in descending order of influence: farmland  $\rightarrow$  forest-grass land  $\rightarrow$  salinized land  $\rightarrow$  water body  $\rightarrow$  others. Moreover, the influence was lower during the wet season than during the dry season.

**339 **3 Discussion**

340 Given the seasonal differences, the unbalanced distribution of precipitation resulted in an apparent 341 variation in the surface runoff and further imbalanced the spatial distribution of the water quality in the 342 research zone (Fan et al., 2012; Prathumratana et al., 2008; Li et al., 2015). During the wet season (May) in 343 the Jinghe Oasis, melted water from mountain ice and snow promote the flow in the Jing River, thereby resulting in a significant increase in surface runoff leading to an improved water quality in the rainy season 344 345 compared to the dry season. During the dry season, the aquatic plants in rivers and lakes grow as the 346 temperature rises, which can, to a certain degree, absorb and purify part of the water quality parameters. 347 Therefore, significant and seasonal changes in the surface runoff at the research zone are important factors resulting in noticeable differences in the spatial distribution of water quality characteristics during the wet 348 349 and dry seasons. Another major factor that causes differences in the spatial distribution of water quality is 350 the change in land use/cover, especially farmland. During the dry season, farmland areas have a greater 351 influence on more water quality variables than they do during the rainy season because of their intensive 352 fertilization and agricultural runoff from soil erosion (Ngoye et al, 2004;Li et al., 2009; Tran et al., 2010). 353 Multiple factors threaten the ecological safety of the Jinghe Oasis system. Especially in recent years, the 354 lakeside desertification zone has rapidly expanded because of the decrease in the Ebinur Lake area and the 355 degradation of lakeside vegetation under influence of the strong winds in Alashankou. In the current 356 overall situation, the human activities influencing land use/cover changes are directly related to the development of the vulnerable ecological area that surrounds Ebinur Lake. 357

358 Recent statistics indicate that the annual growth rate of the population in the Jinghe Oasis is 359 approximately 2.49%, which is slightly higher than previous growth rates (Li, 2006). Under the stress of a large population, the amount of inappropriate activities that negatively impact land use/cover in the Jinghe 360 Oasis will increase. For the last 30 years, cotton has been the major crop in the Jinghe Oasis. The results of 361 362 the current study indicate that the sampling points surrounding the farmland in the research zone have 363 lower water quality values than the other studied regions. The primary livelihoods of the urban residents 364 around the Ebinur Lake area are agricultural and animal husbandry industries. Pollutants that result in high TP and NH3-N contents in water include the excessive application of chemical fertilizers on farmlands, the 365 production of livestock manure in rural villages, randomly stocked garbage, and domestic wastewater. The 366 367 improper application of chemical fertilizers and pesticides to a vast area leads to high water nitrogen and phosphorus contents, resulting in the spread of algae in river sections. Consequently, the amount of 368 369 dissolved oxygen in the river may decrease, the water quality may deteriorate, and eutrophication may 370 occur. Furthermore, this scenario poses a serious salinization problem. Certain measures have been 371 implemented for the ecological protection of Ebinur Lake, such as returning farmland to forest, cultivating 372 ecological forests, and promoting efficient irrigation and water-saving technologies. However, these measures promote the gradual expansion of the lake area and result in different degrees of negative 373 consequences. The most apparent result has been the rise of the underground water level, which has 374 375 aggravated land salinization in the lowland areas and resulted in vast expanses of uncultivated former 376 agricultural lands. Statistics indicate that soil salinization in the Ebinur Lake area mainly occurs in Bortala 377 River, Jing River, the villages and towns surrounding Ebinur Lake, areas downstream of the Daheyanzi

378 River, and areas north of Bole City (Mi et al.,2010). Severe soil salinization has seriously affected the 379 farming of crops; therefore, some farmers have increased the amount of chemical fertilizers they apply to 380 increase yield, which also increases the pollution of the water and soil. Others have even abandoned their 381 land, thereby causing land use/cover change.

Most rivers in Xinjiang are characterized by a low water yield, short flow, small water environmental capacity, poor self-cleaning capability, and low tolerance to pollution. Hence, an artificial change in the land use and exploration of resources in lake regions lead to an evident correlation between land use/cover types and water quality. In addition, the scientific utilization and protection of the water resources of Ebinur Lake and the scientific application of chemical fertilizers and improvement of their application rates are important actions and should be addressed to achieve sustainable development in the agricultural irrigation zones of the Jinghe Oasis and rivers of Xinjiang.

**389 4 Conclusions**

390 The spatial distribution characteristics of water quality in the Jinghe Oasis and their correlation with 391 land use/cover types were analysed, and the following conclusions were drawn.

392 (1) Using the SOM method based on non-hierarchical k-means classification, 47 water quality 393 sampling points were divided into 6 types, and the time sequence characteristics of the research zone were 394 better recognized in the classification results. Clusters 1 to 3 comprised samples from the wet season (May 395 2015), whereas Clusters 4 to 6 comprised monitoring samples from the dry season (October 2015). In 396 general, the COD, SS, NUT, TN, and NH3-N contents were high. The SD value was high in Clusters 1, 4, 397 and 6. In addition, high BOD and TP values were mainly concentrated in Clusters 4 and 6. Based on these 398 findings, the water quality at different layers of the research zone was further evaluated. The results show 399 that Clusters 1 to 6 do not satisfy potable water quality standards.

400 (2) The correlations between the land use/cover types and water quality parameters for Clusters 1 to 6 401 were analysed according to the hierarchical results of the water quality parameters. The comprehensive 402 analysis indicates that the farmland, forest–grass land, and salinized land exerted significant influences on 403 the water quality parameters of the Jinghe Oasis. In Clusters 1, 2, and 6, the size of the water area, to a 404 certain extent, also influenced changes in the water quality parameters.

405 (3) During the wet and dry seasons, the influences that various land use/cover types in the research 406 zone had on the water quality parameters exhibited the following order, in descending order of influence: 407 farmland  $\rightarrow$  forest-grass land  $\rightarrow$  salinized land  $\rightarrow$  water body  $\rightarrow$  others. Moreover, the influences were 408 lower during the wet season than during the dry season.

In general, the land use/cover type, area percentage, and water quality in the Jinghe Oasis demonstrated apparent correlations. The results of this study can tentatively explain the relationship between water quality and land use/cover types in different clusters by the SOM method. This work provides new insight for further studies on the correlation between land use/cover and water quality in the Jinghe Oasis, as well as a scientific reference for formulating regulations and control policies for the spatial development and water environmental protection of the Jinghe Oasis.

415 **Acknowledgements:** Thanks to the National Meteorological Information Center data provided 416 meteorological data. The research was carried out with the financial support provided by the Natural 417 Science Foundation of Xinjiang Uygur Autonomous Region, China (2016D01C029); National Natural

- 418 Science Foundation of China (41361045), the Scientific and technological talent training program of
- 419 Xinjiang Uygur Autonomous Region (grant No. QN2016JQ0041). The authors wish to thank the referees
- 420 for providing helpful suggestions to improve this manuscript.

**421 **References**

[revised manuscript text omitted]

**Figure captions**